# Characterising the genetic architecture of changes in adiposity during adulthood using electronic health records

Samvida S. Venkatesh [1,2] ✉, Habib Ganjgahi[2,3], Duncan S. Palmer [2,4], Kayesha Coley [5], Gregorio V. Linchangco Jr.[6,7], Qin Hui [6,7], Peter Wilson[7,8], Yuk-Lam Ho [9], Kelly Cho[9,10], Kadri Arumäe[11], Million Veteran Program*, Estonian Biobank Research Team*, Laura B. L. Wittemans[12,13], Christoffer Nellåker [2,13], Uku Vainik[11,14,15], Yan V. Sun [6,7], Chris Holmes[3,16,17], Cecilia M. Lindgren[1,2,13,18] ✉ & George Nicholson [3] ✉

Obesity is a heritable disease, characterised by excess adiposity that is measured by body mass index (BMI). While over 1,000 genetic loci are associated with BMI, less is known about the genetic contribution to adiposity trajectories over adulthood. We derive adiposity-change phenotypes from 24.5 million primary-care health records in over 740,000 individuals in the UK Biobank, Million Veteran Program USA, and Estonian Biobank, to discover and validate the genetic architecture of adiposity trajectories. Using multiple BMI measurements over time increases power to identify genetic factors affecting baseline BMI by 14%. In the largest reported genome-wide study of adiposity-change in adulthood, we identify novel associations with BMI-change at six independent loci, including rs429358 (*APOE* missense variant). The SNP-based heritability of BMI-change (1.98%) is 9-fold lower than that of BMI. The modest genetic correlation between BMI-change and BMI (45.2%) indicates that genetic studies of longitudinal trajectories could uncover novel biology of quantitative traits in adulthood.

Obesity, the accumulation of excess body fat[1], which is associated with increased disease burden[2,3], has a strong genetic component[4]. The heritability of body mass index (BMI) is estimated to be 40–70%[4–6], and genome-wide association studies (GWASs) have implicated over 1000 independent loci associated with a range of obesity traits[4]. The dynamic process of change in weight over time is also thought to have a genetic component[7,8]. Recent studies reveal the shifting genetic landscape of infant, childhood, and adolescent BMI, which detect age-specific transient effects by performing age-stratified GWASs[9–11]. Adult twin studies[12–14] and an electronic health record (EHR)-based population study[15] indicate that long-term patterns of change in adiposity are heritable and have a distinct genetic component to baseline obesity levels. However, less is known about the specific variants and genes

that contribute to patterns of adulthood adiposity change. This paucity of GWASs of long-term trajectories of weight change can be partially attributed to the challenges in building and maintaining large-scale genetics cohorts that follow participants over their lifetime[16].

Longitudinal data are a key feature of EHRs, whose increased adoption in the clinic and integration into biobanks has powered cost-efficient and scalable genetics research[17,18]. Despite biases in EHR data, including sparsity, non-random missingness, data inaccuracies, and informed presence, EHR-based genetics studies reliably replicate results from purpose-built cohorts[19–21]. Recent advances in the extraction of phenotypes from longitudinal EHRs at scale show that, as expected[22,23], the mean of repeat quantitative measurements can outperform cross-sectional phenotypes for genetic discovery[24,25].

A full list of affiliations appears at the end of the paper. *Lists of authors and their affiliations appear at the end of the paper. ✉e-mail: samvida@well.ox.ac.uk; cecilia.lindgren@bdi.ox.ac.uk; george.nicholson@stats.ox.ac.uk

Repeat measurements further allow for the estimation of longitudinal metrics of trait change, such as trajectory-based clusters[26], linear slope[27], and within-individual variability over time[28], all of which may provide additional information to uncover the genetic underpinnings of disease.

A variety of approaches are available for harnessing the longitudinal component of trajectories in EHR data. Simple models target the gradient of a linear fit over time, such as in a longitudinal linear mixed-effects model framework[28–30]. More complex regression modelling approaches are employed to investigate non-linear changes over time. For example, semi-parametric regression models[31] generate flexible longitudinal patterns from combinations of basis functions, such as B-splines, regularised to induce a suitable degree of temporal smoothness[32–35]. Subgroups of individuals with similar non-linear trajectories are often identified through clustering approaches, with subgroup membership then tested for association with clinical outcomes or genetic variation[36–41]. Although it is possible to fit full joint models that incorporate both genetic data and longitudinal trajectories simultaneously[28], two-stage approaches wherein summary metrics from models of longitudinal EHRs are taken forward for genetic association analyses are popular for their computational efficiency[27].

In this study, we leveraged longitudinal EHRs linked to the UK Biobank (UKBB)[42], Million Veteran Program (MVP)[43,44], and Estonian Biobank (EstBB)[45] to study the genetic architecture of change in adiposity over adulthood. We developed a two-stage analytical pipeline, utilising statistical methods with a history of application in the EHR data context, to derive linear and non-linear trajectories of BMI and weight over time, and to identify clusters of individuals with similar adiposity trajectories. In the second stage, we carried forward the latent phenotypes from these models, which capture both baseline obesity trait levels and change in obesity traits over time, to perform the largest reported genome-wide association analyses for adiposity change in adulthood. Our results demonstrate the added value of EHR-derived longitudinal phenotypes for genetic discovery.

## Results

### Longitudinal data help identify novel genetic signals for obesity

We obtained BMI and weight records for up to 177,098 individuals of white–British ancestry with up to 1.48 million measurements in UKBB longitudinal records from general practitioner (GP) and UKBB assessment centre measurements (Table 1 and Supplementary Fig. 3). For each individual, we estimated linear change in BMI or weight over time using a linear mixed-effects (LME) model with random intercepts and random longitudinal gradients (Fig. 1A) within six strata—defined as the pair-wise combinations of two adiposity traits (BMI, weight) with three sex subsets (women-only, men-only, combined sexes). We sought replication of genetic findings in two external cohorts with longitudinal EHR data—MVP ($N = 437{,}703$) and EstBB ($N = 127{,}769$)—whose demographic and obesity trait characteristics are distinct from UKBB. Individuals in MVP are predominantly male (92.4%) and on average 3.5 units of BMI heavier than male participants in the UKBB; on the other

hand, participants in EstBB are of similar BMI to those in the UKBB, but are on average 6–8 years younger than their UKBB counterparts (Supplementary Data 23).

We first investigated whether the individual-level random-intercept terms outputted by the longitudinal LME model, by sharing information across multiple BMI measurements, provided higher statistical power for GWAS than one based on a single, cross-sectional BMI measurement per individual. Despite our GWAS being 4-fold smaller than the largest published analyses[46], we identify 14 novel loci and refine 53 previously described signals for obesity traits among the 374 unique fine-mapped lead single-nucleotide polymorphisms (SNPs) ($P < 5 \times 10^{-8}$) across all strata (Fig. 2A, Supplementary Fig. 13, and Supplementary Data 2), see Methods for conditional analysis to classify novel, refined, and reported SNPs[47]. The 53 refined SNPs are conditionally independent of and represent stronger associations ($P < 0.05$) than published SNPs in this population. Together, the refined and novel SNPs explain 0.33% of variance in baseline BMI (in addition to the 2.7% explained by previously published SNPs), and 0.83% of variance in baseline weight (in addition to the 4.7% explained by previously reported SNPs) (Fig. 2B). We further quantified the power gained from estimating baseline BMI over repeat longitudinal measurements per individual by comparing genome-wide significant (GWS) SNPs from our baseline BMI GWAS to the largest published BMI meta-analysis to date[46]. We observe an increase in median chi-squared statistics of GWS SNPs from either study of between 13.4% (females) to 14.8% (males) in our GWAS over what would be expected from a cross-sectional GWAS of equivalent sample size.

Nine of the 14 novel SNPs replicate at $P < 3.6 \times 10^{-3}$ (family-wise error rate (FWER) controlled at 5% across 14 tests using the Bonferroni method) in at least one of (1) baseline obesity estimated with LME model intercepts in up to 437,703 individuals the MVP cohort, (2) baseline obesity estimated with LME model intercepts in up to 125,209 individuals the EstBB cohort, or (3) UKBB assessment centre measurements of cross-sectional obesity in up to 230,861 individuals not included in the discovery GWAS (Supplementary Data 3). These include rs6769383, whose nearest gene *EDEM1* is involved in carbohydrate metabolism[48], rs2861761, whose nearest gene *TENM2* is enriched in white adipocytes[49], rs11156978 whose nearest gene *CHD8* is associated with impaired glucose tolerance in mouse knockouts[50], and rs7962636, whose nearest gene *MED13L* is a transcriptional regulator of white adipocyte differentiation[51]. We also replicate in MVP the male-specific BMI association of rs79586444, whose nearest gene, *DUSP26,* is associated with decreased high-density lipoprotein (HDL) cholesterol in mouse knockouts[52].

Intra-individual variance is another longitudinal metric of interest, however we (Supplementary Fig. 15) and others[28] find no genetic variants associated with intra-individual variance in weight over time. While the intra-individual mean and baseline trait modelled from LME are phenotypically ($R^2 > 0.95$) and genetically highly correlated ($R^2 > 0.99$) (Supplementary Fig. 17), the LME intercept appears better powered for genetic association testing than the average trait, as we

**Table 1 | Characterisation of obesity trait data in longitudinal records curated from UK Biobank assessment centre visits and linked general practitioner (GP) records**

| Trait | Sex | Number of individuals | Number of obs. | Mean number of repeat obs. (SD) | Mean length of follow-up, years (SD) | Mean age at first obs., years (SD) | Median trait value at first obs. (IQR) |
|---|---|---|---|---|---|---|---|
| BMI, kg/m² | F | 88,243 (54.4%) | 696,984 | 7.90 (7.34) | 13.7 (6.63) | 48.6 (9.68) | 24.6 (22.2, 27.9) |
| BMI, kg/m² | M | 73,965 (45.6%) | 581,161 | 7.86 (7.12) | 12.8 (6.55) | 50.1 (9.59) | 26.1 (24.0, 28.7) |
| Weight, kg | F | 96,625 (54.6%) | 816,885 | 8.45 (8.33) | 13.9 (6.62) | 48.3 (9.63) | 65.0 (59.0, 74.0) |
| Weight, kg | M | 80,473 (45.4%) | 666,258 | 8.28 (7.82) | 12.9 (6.57) | 50.0 (9.57) | 81.6 (73.8, 90.0) |

*BMI* body mass index, *obs.* observation, *S.D.* standard deviation, *I.Q.R.* inter-quartile range.

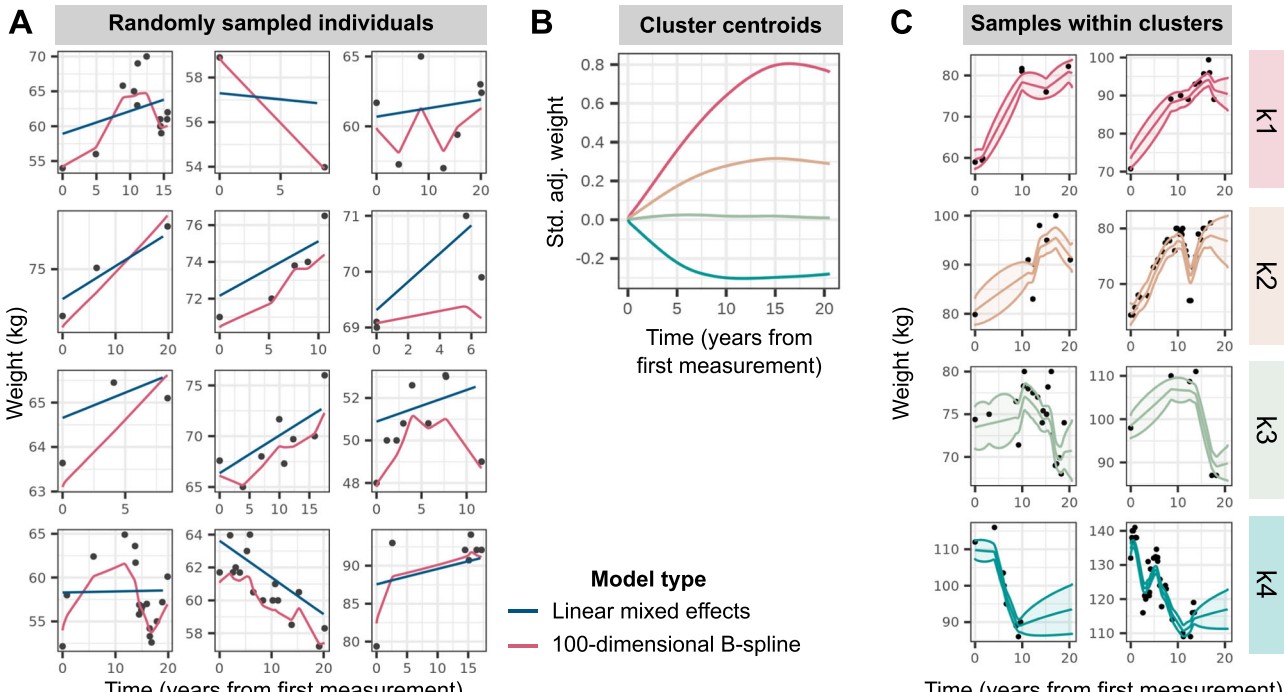

**Fig. 1 | Modelling of longitudinal obesity trait trajectories. A** Weight trajectories over time, measured as years from the first measurement, in a random sample of 12 individuals in the sex-combined strata. Black points display observed weight records, with blue and pink lines representing predicted fits from linear mixed-effects models and regularised high-dimensional spline models respectively. **B** Trajectories of cluster centroids, plotted as standardised (std.) and covariate-adjusted (adj.) weight over time (years from first measurement), for the four clusters determined via partitioning-around-medoids (PAM) clustering with a customised distance matrix (see Methods) constructed from the high-dimensional B-spline coefficients estimated in **A. C** Weight trajectories over time for a random sample of individuals in the 99th percentile probability of belonging to each cluster, as determined by parametric bootstrap. The lines display predicted fits and ribbons represent 95% confidence intervals around the mean fit.

discover up to 1.2× more GWS variants associated with the former (Supplementary Data 20).

Ascertainment bias in our discovery cohort could arise from the over-representation of heavier participants in EHR data (Supplementary Data 4)[53]. On average, women with ten or more weight measurements are 8.3 kg (3.7 units of BMI) heavier than their counterparts with 1–3 measurements; for men, this is an 8.2 kg (3.1 units of BMI) difference. However, the BMI-intercept metric from our longitudinal data is genetically perfectly correlated with the un-ascertained cross-sectional BMI in Genetic Investigation of ANthropometric Traits (GIANT) 2019[46] ($r_G = 1$ and $P < 1 \times 10^{-16}$ in all strata), and 96% of the GWS associations ($P < 5 \times 10^{-8}$) identified in our GWAS have either been reported, or are correlated with reported obesity-associated SNPs in the GWAS Catalog[54] (Supplementary Data 1).

### *APOE* variant associated with weight loss over time, independent of baseline obesity

To identify genetic variants that affect change in adiposity over time, we performed GWASs for patterns of BMI and weight change adjusted for baseline measurements, defined in two ways. First, we created a linear phenotype from subject-specific random gradients, estimated within the LME model framework. Second, to capture non-linear patterns of temporal change, we modelled longitudinal variation in obesity traits using a regularised high-dimensional B-spline basis[31] (Fig. 1). Within each of the six strata, we identified four clusters of individuals using *k*-medoids clustering[55,56], representing high gain (k1), moderate gain (k2), stable (k3), and loss (k4) trajectories, and estimated each individual's probability of belonging to a cluster based on their posterior non-linear obesity trait trajectory (Fig. 1 and Supplementary Fig. 5). We performed GWASs on the linear slope-change phenotype and on individuals' logit-transformed posterior probabilities of

membership in the high gain cluster (k1), high and moderate gain clusters (k1 and k2), or all but the loss cluster (k1, k2, and k3). All analyses were adjusted for baseline obesity trait and confounders, including length of follow-up and number of follow-up measures, to mitigate survivor bias.

A common missense variant in *APOE* (rs429358) is associated with decrease in both BMI and weight over time, and lower posterior probabilities of gain-cluster membership in all analysis strata (Table 2). Each copy of the minor C allele of rs429358 (minor allele frequency (MAF) = 0.16) is associated with 0.060 standard deviation (SD) decrease (95% confidence interval (CI) = 0.050–0.069, $P = 8.6 \times 10^{-35}$) in expected BMI slope over time and 0.063 SD decrease (0.054–0.072, $P = 6.0 \times 10^{-42}$) in expected weight slope over time (Fig. 3A). Independent of baseline obesity, carriers of the minor C allele of rs429358 are at lower odds of membership in the high-gain BMI and weight clusters (odds ratio (OR) = 0.976, 95% CI = 0.97–0.98, $P < 4.9 \times 10^{-19}$), lowering the membership posterior probability from 40% to 39% on average (Fig. 3B). Although the minor allele of rs429358 is also associated with lower baseline BMI ($\beta = 0.015$ SD lower BMI-intercept, 95% CI = 0.0054–0.024) and weight ($\beta = 0.011$ SD lower weight intercept, 95% CI = 0.0029–0.020), these associations do not reach GWS ($P > 0.002$).

The association of rs429358 with adiposity-change phenotypes was replicated at $P < 1.39 \times 10^{-3}$ (FWER controlled at 5% across six variants and six traits tested) in: (1) up to 437,703 individuals in the MVP cohort, (2) up to 125,209 individuals in the EstBB, and (3) up to 17,035 individuals in UKBB with multiple measurements of weight and BMI at repeat assessment centre visits who were excluded from the discovery analyses (Fig. 4 and Supplementary Data 5). Further, based on 301,943 UKBB participants who were not included in the discovery GWASs, and who reported weight change in the last year as

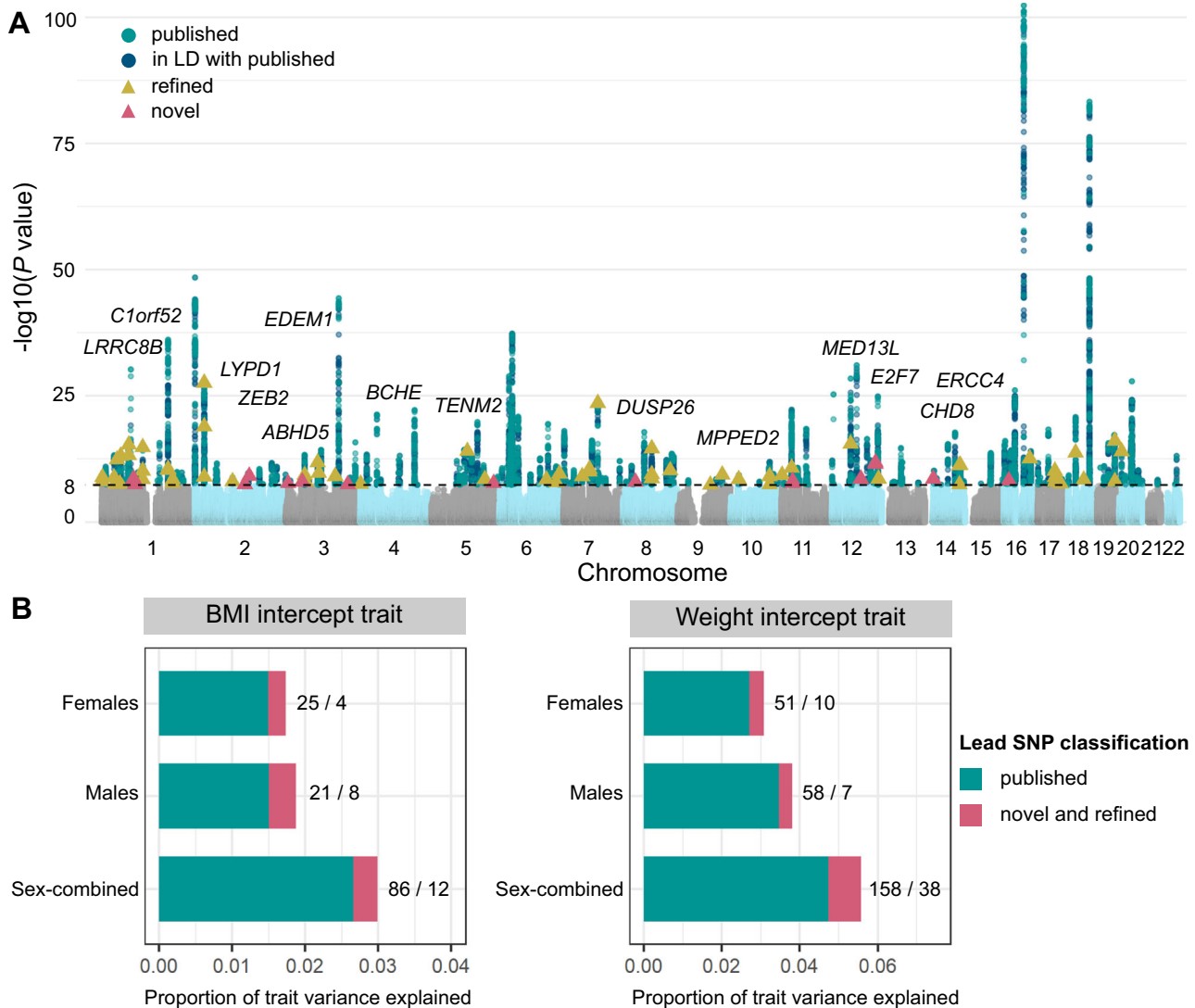

**Fig. 2 | Genome-wide novel and refined SNP associations with baseline obesity estimated over the measurement window for each individual. A** Combined Manhattan plot displaying genome-wide SNP associations estimated using linear mixed-model tests in `BOLT-LMM`[84] with obesity trait (BMI or weight) across female, male, and sex-combined analysis strata. Each point represents an SNP, with genome-wide significant (GWS) SNPs ($P < 5 \times 10^{-8}$) coloured in green for previously published obesity associations, blue for SNPs in LD ($r^2 > 0.1$) with published associations, yellow for refined SNPs that represent conditionally independent ($P_{conditional} < 0.05$) and stronger associations with baseline obesity than published SNPs in the region, and pink for novel associations (see Methods[47]). Novel SNPs are annotated to their nearest gene. **B** Proportion of variance in baseline BMI and weight that can be explained by the fine-mapped independent lead SNPs in each strata. In green is the proportion of variance explained by previously published obesity-associated variants (and those in LD with these variants), while that explained by novel and refined variants is in pink. The numbers represent the number of lead SNPs in each of these categories (published/refined and novel).

"gain", "about the same", or "loss", we found that carriers of each additional copy of the minor C allele of rs429358 are at 0.956 (95% CI = 0.94–0.97) lower odds of being in a higher ordinal weight-gain category, independent of their BMI (Fig. 3C and Supplementary Data 6). We observe consistent effect direction of the rs429358 association with both estimated and self-reported weight loss over time in individuals who self-identify as Asian (maximum N = 8324 individuals), Black (6796), mixed (2681), white not in the white–British ancestry subset (47,174), and other (3994) ethnicities (see Methods for ancestral group definitions, Supplementary Fig. 1 and Supplementary Data 7).

Finally, we tested for the effect of rs429358 on change in abdominal adiposity in up to 44,154 individuals of white–British ancestry in UKBB who were not in the discovery set, with repeated assessment centre measurements of waist circumference (WC) and waist-to-hip ratio (WHR). Each copy of the C allele is associated with 0.040 SD decrease (95% CI = 0.021-0.049, $P = 2.3 \times 10^{-5}$) in

expected WC slope over time and 0.031 SD decrease (0.012–0.050, $P = 1.1 \times 10^{-3}$) in expected WHR slope over time, independent of baseline values (Fig. 3D and Supplementary Data 6). While the effect direction remains consistent, these associations are no longer significant upon adjustment for BMI (all $P > 0.1$), suggesting that the observed loss in abdominal adiposity over time may represent a reduction in overall adiposity.

We additionally performed a longitudinal phenome-wide scan to test for the association of rs429358 with changes in 45 quantitative biomarkers obtained from the UKBB-linked primary care records. Each copy of the C allele is associated with an increase in expected slope change over time of total cholesterol ($\beta = 0.030$ SD increase, $P = 6.4 \times 10^{-12}$), C-reactive protein (CRP) ($\beta = 0.026$, $P = 9.6 \times 10^{-7}$), and HDL cholesterol ($\beta = 0.022$, $P = 1.0 \times 10^{-5}$), but a decrease in expected slope change over time of triglycerides ($\beta = -0.027$, $P = 2.7 \times 10^{-7}$), potassium ($\beta = -0.023$, $P = 3.9 \times 10^{-6}$), lymphocytes ($\beta = -0.020$, $P = 4.0 \times 10^{-5}$), and haemoglobin concentration ($\beta = -0.016$, $P = 1.0 \times 10^{-3}$) (FWER controlled at 5%

**Table 2 | Lead SNPs identified from genome-wide association studies (GWAS) of posterior probability of membership in an adiposity-change cluster (high gain k1, high/moderate gain k1/k2, or high/moderate gain k1/k2/k3 and steady k1/k2/k3), independent of baseline obesity**

| Trait | SNP | chr:pos (hg37) | MAF | Nearest TSS | Adiposity-change cluster | Sex-combined OR (95% CI) | Sex-combined P value | Female OR (95% CI) | Female P value | Male OR (95% CI) | Male P value | Sex-heterogeneity P value |
|---|---|---|---|---|---|---|---|---|---|---|---|---|
| Weight | rs9467663 | 6:2602456 | 0.418 | H4C1 | k1 | 1.01 (1.01-1.01) | 1.6e-09 | 1.01 (1.01-1.02) | 4e-06 | 1.01 (1-1.01) | 0.00081 | 0.785 |
| BMI | chr6:26076446 | 6:26076446 | 0.437 | HFE | k1 | 1.01 (1.01-1.02) | 2.1e-09 | 1.01 (1.01-1.02) | 7.1e-06 | 1.01 (1.01-1.02) | 2.7e-05 | 0.478 |
| BMI | rs11778922* | 8:20493349 | 0.379 | LZTS1 | k1 | 0.99 (0.987-0.994) | 2.1e-06 | 0.984 (0.979-0.99) | 1.3e-08 | 0.998 (0.992-1) | 0.41 | 0.000578 |
| BMI | rs61955499* | 13:112565161 | 0.012 | SOX1 | k1, k2, or k3 | 0.978 (0.961-0.995) | 0.012 | 0.935 (0.913-0.958) | 3.4e-08 | 1 (0.978-1.03) | 0.78 | 4.71e-05 |
| Weight | rs12953815* | 18:2586907 | 0.496 | NDC80 | k1, k2, or k3 | 1 (1-1.01) | 0.02 | 1 (0.995-1) | 0.97 | 1.01 (1.01-1.02) | 1.7e-08 | 2.03e-05 |
| BMI | rs429358 | 19:45411941 | 0.156 | APOE | k1 | 1.02 (1.02-1.03) | 4.9e-19 | 1.03 (1.02-1.03) | 3e-12 | 1.02 (1.01-1.03) | 5.2e-08 | 0.764 |
| BMI | rs429358 | 19:45411941 | 0.156 | APOE | k1 or k2 | 1.02 (1.02-1.03) | 1.1e-17 | 1.02 (1.02-1.03) | 2.6e-11 | 1.02 (1.01-1.03) | 5.4e-08 | 0.689 |
| BMI | rs429358 | 19:45411941 | 0.156 | APOE | k1, k2, or k3 | 1.02 (1.02-1.03) | 2.5e-16 | 1.02 (1.02-1.03) | 9.3e-11 | 1.02 (1.01-1.03) | 2.5e-07 | 0.714 |
| Weight | rs429358 | 19:45411941 | 0.156 | APOE | k1 | 1.02 (1.02-1.03) | 1.8e-20 | 1.02 (1.02-1.03) | 1.5e-11 | 1.02 (1.01-1.03) | 8.9e-08 | 0.807 |
| Weight | rs429358 | 19:45411941 | 0.156 | APOE | k1 or k2 | 1.02 (1.02-1.03) | 2.7e-19 | 1.02 (1.02-1.03) | 6.6e-12 | 1.02 (1.01-1.02) | 6.8e-07 | 0.886 |
| Weight | rs429358 | 19:45411941 | 0.156 | APOE | k1, k2, or k3 | 1.02 (1.02-1.03) | 1.8e-17 | 1.02 (1.02-1.03) | 1e-10 | 1.02 (1.01-1.02) | 4.3e-06 | 0.875 |

Variants marked with * have $5 \times 10^{-9} >$ P value $> 1 \times 10^{-8}$. MAF minor allele frequency (European-ancestry), TSS transcription start site, SE standard error, OR odds ratio, CI confidence interval.

across 45 tests via the Bonferroni method) (Fig. 3E and Supplementary Data 8).

The *APOE* locus is a highly pleiotropic region that is associated with lipid levels[57,58], Alzheimer's disease[59,60], and lifespan[61,62], among other traits[63], both in the UKBB (Supplementary Fig. 14) and elsewhere. Excluding the 242 individuals with diagnoses of dementia or Alzheimer's disease in our replication datasets did not alter associations of rs429358 with any of the longitudinal obesity traits (Supplementary Fig. 2), indicating that they are unlikely to be driven solely by weight loss that accompanies dementia. Despite the association of rs429358 with lifespan, we found no association between this variant and follow-up metrics in our study (Supplementary Data 22); we also found no significant difference in the effect of this variant on adiposity change from two sets of models: (1) without including age and related covariates, i.e., follow-up metrics and year of birth, and (2) with these covariates (heterogeneity P value $P_{het} > 0.05$) (Supplementary Fig. 16). Finally, we observe no associations between 135 of 138 published lifespan-associated genetic variants and our adiposity-change phenotypes at $P < 3.6 \times 10^{-4}$ (FWER controlled at 5% across 138 tests via the Bonferroni method). Of the three SNPs associated with both weight change and lifespan, two (rs429358 and rs7412) are variants in the *APOE* gene, and rs1085251 is a known obesity association in the *FTO* locus (Supplementary Data 16).

## Genome-wide architecture of change in adiposity over time is distinct from baseline adiposity

We identify six independent genetic loci associated with distinct long-itudinal trajectories of obesity traits (Table 2). This included the *APOE* locus above and five signals in intergenic regions. rs9467663 (OR = 1.011 for membership in the high-gain weight cluster, $P = 1.6 \times 10^{-9}$) and chr6:26076446 (OR = 1.012 for membership in the high-gain BMI cluster, $P = 2.1 \times 10^{-9}$), are reported associations with haematological traits[64]. We identify two SNPs, rs11778922 and rs61955499, with female-specific effects on BMI change. rs11778922 (OR = 0.984 for membership in the high-gain BMI cluster, $P = 1.3 \times 10^{-8}$, sex-heterogeneity $P_{sexhet} = 5.8 \times 10^{-4}$, see Methods) has previously been nominally associated with BMI in females[46], and rs61955499 (OR = 1.070 for membership in the BMI loss cluster, $P = 3.4 \times 10^{-8}$, $P_{sexhet} = 4.7 \times 10^{-5}$), has previously been nominally associated with low-density lipoprotein (LDL) cholesterol levels[65]. Finally, rs12953815 is associated with male-specific weight change (OR = 1.012 for membership in the weight loss cluster, $P = 1.7 \times 10^{-8}$, $P_{sexhet} = 2.0 \times 10^{-5}$) and has been previously nominally associated with lung function[66].

Other than rs429358, none of the lead variants for adiposity change replicated in either MVP or EstBB at $P > 1.39 \times 10^{-3}$ (FWER controlled at 5% across 6 variants via the Bonferroni method) (Supplementary Data 5). However, we were only sufficiently powered to replicate the effects of three of these in MVP (rs9467663, chr6:26076446, and the male-specific variant rs12953815), and none in EstBB, as replication at 80% power required sample sizes of between 116,000 to 234,000 individuals with repeat measurements of BMI (Supplementary Data 25).

While all lead variants in the discovery GWASs remain significant at $P < 5 \times 10^{-7}$ in GWASs that are not adjusted for follow-up metrics, we discover three variants in the *FTO* locus that are associated with BMI or weight gain only in analyses that are unadjusted for follow-up metrics (Supplementary Data 21). These associations may reflect genetic contributions to baseline weight rather than weight change, as *FTO* is among the strongest known loci for obesity, and follow-up metrics are strongly positively correlated with baseline obesity (Supplementary Data 4).

The smaller number of independent GWS associations with adiposity change: six, compared to 374 unique lead SNPs associated with baseline obesity traits, is expected given the 7- to 9-fold lower heritability of adiposity change. The heritability explained by

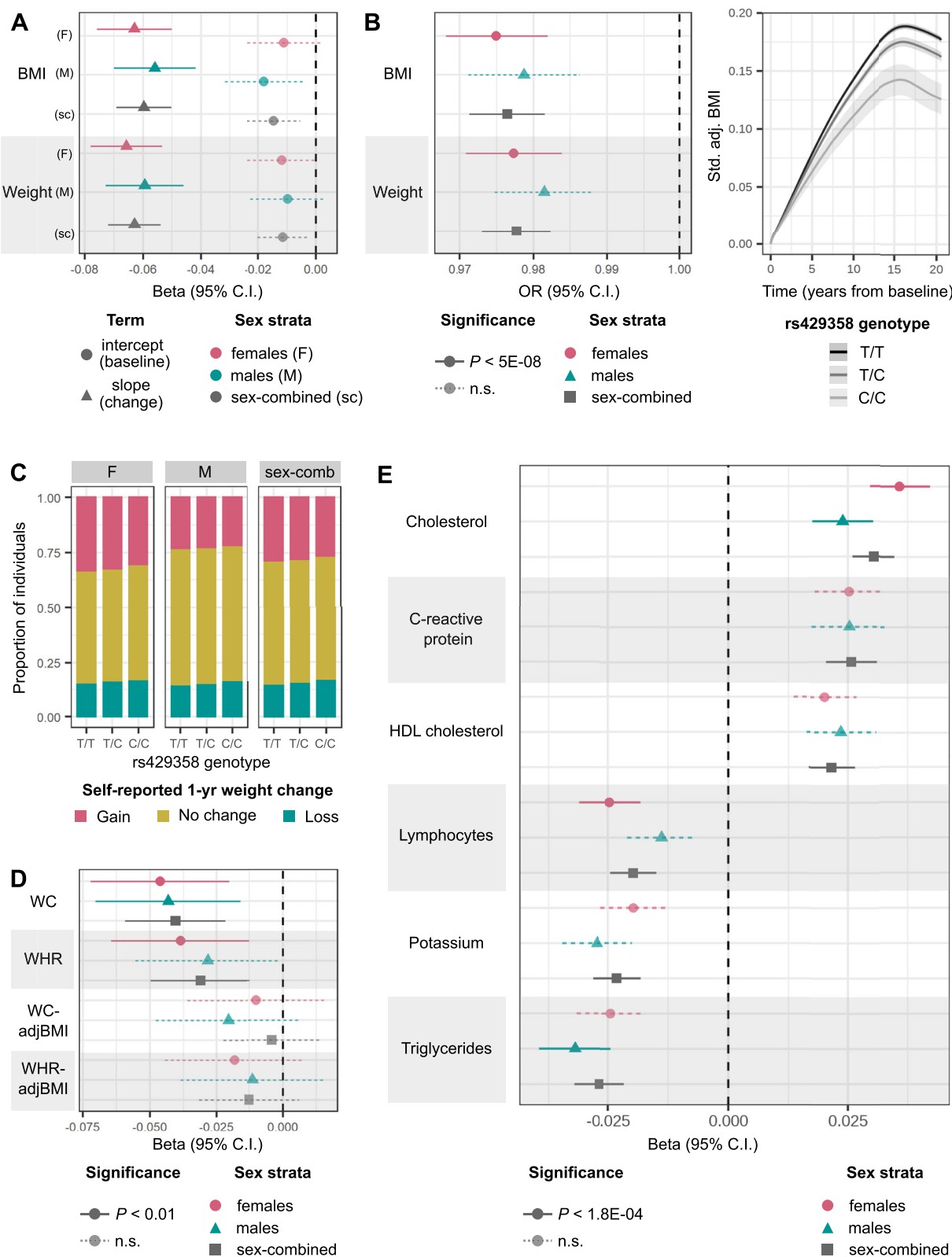

genotyped SNPs ($h_G^2$)[67] of the posterior probability of belonging to an adiposity-gain cluster is between 1.38% (standard error (SE) = 0.53) in men to 2.82% (0.59) in women, while the $h_G^2$ of baseline obesity traits varies between 21.6% (1.09) to 29.0% (1.72) across strata (Fig. 5). Furthermore, we observe that the heritability of BMI and weight trajectories are higher in women than in men (2.89% (0.56) vs 1.05% (0.59) for BMI slopes, $P_{sexhet}$ = 0.012; and 3.42% (0.53) vs 1.69% (0.52) for weight slopes, $P_{sexhet}$ = 9.9 × 10$^{-3}$). Similarly, we estimate the heritability of BMI slopes in the EstBB to be higher in women (2.15% (0.56) in women vs 1.80% (0.98) in men); however, these values are low and must be interpreted with caution. We do not observe a corresponding difference in the $h_G^2$ of baseline BMI or weight between the sexes ($P_{sexhet}$ > 0.1). Finally, baseline and change in obesity traits are genetically correlated, with $r_G$ ranging from 0.35 (95% CI = 0.24−0.45) for

**Fig. 3 | Association of minor C allele of rs429358, missense variant in *APOE*, with various longitudinal phenotypes. A** Mean effect size (beta) and 95% CI for associations of rs429358 with BMI and weight intercepts or linear slope change over time estimated from GWAS in all analysis strata (BMI *N* = 87,908 females and 73,656 males; weight *N* = 96,264 females and 80,144 males). **B** Left: mean OR and 95% CI estimated from GWAS for association of rs429358 with posterior probability of membership in the BMI and weight high-gain clusters (k1). BMI *N* = 87,908 females and 73,656 males; weight *N* = 96,264 females and 80,144 males. Right: modelled trajectories of standardised (std.) covariate-adjusted (adj.) BMI in carriers of the different rs429358 genotypes. **C** Proportion of individuals who self-report weight gain, weight loss, or no change in weight over the past year for carriers of each rs429358 genotype. **D** Mean effect size and 95% CI for associations of rs429358 with

slopes over time of waist circumference (WC) (*N* = 22,680 females and 21,474 males), WC adjusted for BMI (WCadjBMI) (*N* = 22,591 females and 21,379 males), waist-to-hip ratio (WHR) (*N* = 22,677 females and 21,474 males), and WHRadjBMI (*N* = 22,589 females and 21,379 males), estimated from linear mixed-effects models in individuals held-out of discovery analyses (see Supplementary Data 6 for effect estimates and *P* values). **E** Mean effect size and 95% CI for associations of rs429358 with linear slope change in quantitative biomarkers over time, estimated from linear mixed-effects models (*N* between 52,462–146,098 for different biomarkers, see Supplementary Data 8 for details). Across all panels, estimates of trait change are adjusted for baseline trait values, and *P* values for significance are controlled at 5% across number of tests performed via the Bonferroni method. n.s. non-significant.

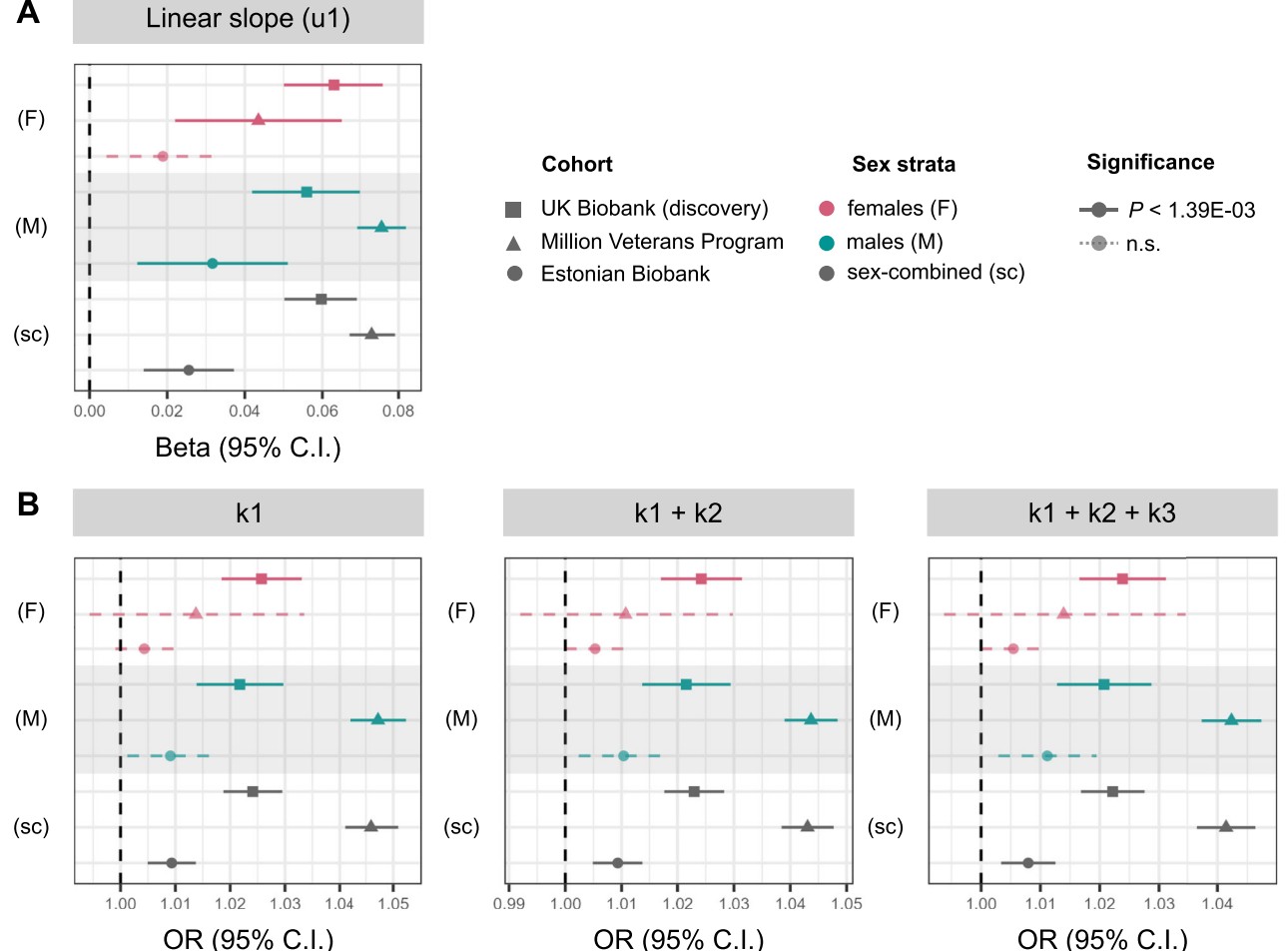

**Fig. 4 | Effect sizes of rs429358 on BMI-change phenotypes in discovery (UK Biobank (UKBB)) and replication (Million Veterans Program (MVP) and Estonian Biobank (EstBB)) datasets. A** Mean effect size (beta) and 95% CI for associations of rs429358 with BMI linear slope change over time estimated from linear mixed-effects models (u1) GWAS in all analysis strata (see Supplementary Data 5 for effect estimates and *P* values). **B** Mean OR and 95% CI for association of all obesity-change lead variants with posterior probability of membership in the BMI high-gain

cluster (k1), high or moderate gain clusters (k1 + k2), or all but loss clusters (k1 + k2 + k3). Across all panels, UKBB *N* = 162,208, MVP *N* = 437,703, EstBB *N* = 127,760; see Supplementary Data 23 for sex-stratified sample sizes. All estimates of trait change are adjusted for baseline trait values, and *P* values for significance are controlled at 5% across number of tests performed via the Bonferroni method. n.s. non-significant.

weight in women to 0.91 (0.59–1.23) for BMI in men (Fig. 5). As expected given their positive correlation, we observe inflation of the $\chi^2$ statistics for adiposity-change slope associations amongst lead variants for baseline adiposity (Supplementary Fig. 19). While the genetic correlation between baseline adiposity and adiposity change appears to be higher in men as compared to women, these estimates have wide CIs (overlapping 1) and $P_{sexhet} > 0.05$ for both BMI and weight.

Throughout this study, we evaluate both BMI and weight as obesity traits, and expect these to track closely in adults as height does not change significantly over time. In the 161,891 individuals in our discovery strata with multiple measurements of both BMI and weight, there is a strong correlation between the slopes for weight and BMI change ($r^2 = 0.88$) and between the posterior probabilities of membership in the BMI-gain and weight-gain clusters ($r^2 = 0.73$)

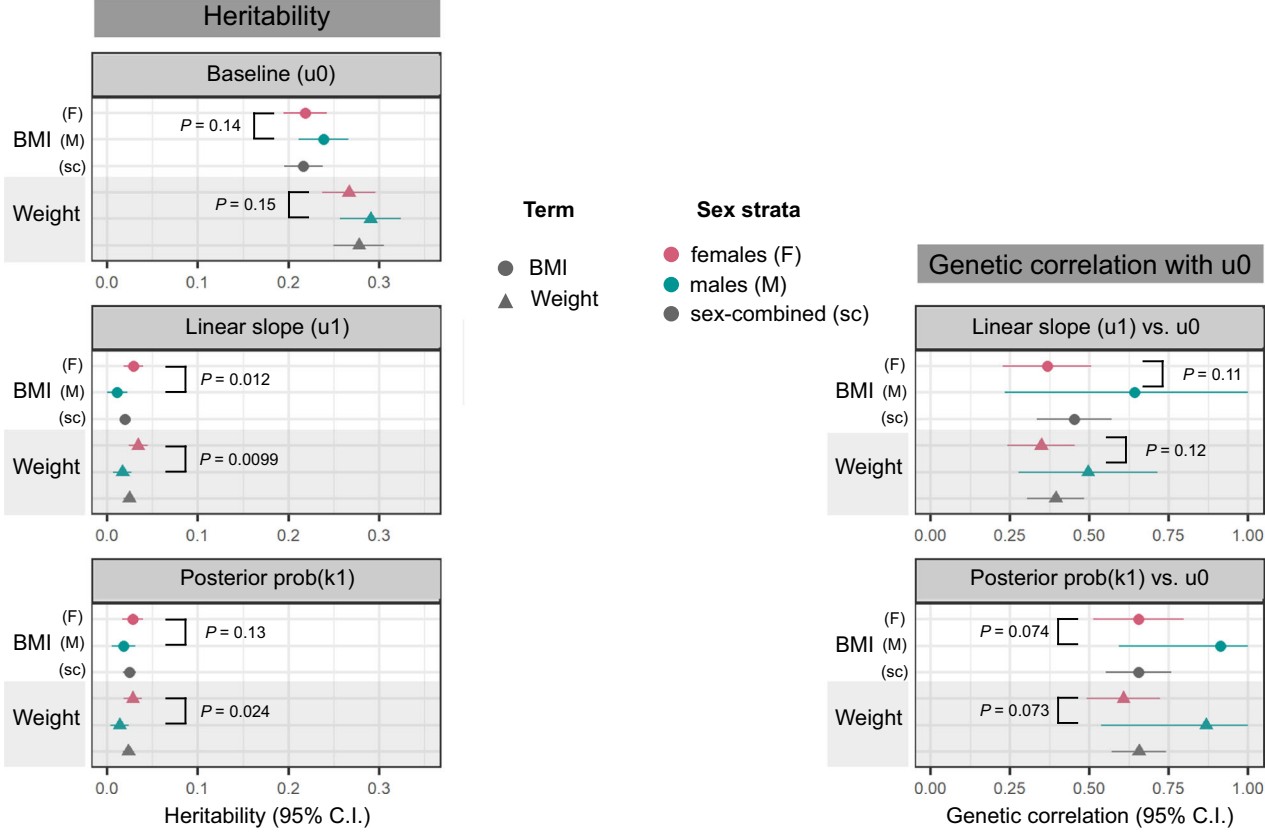

**Fig. 5 | Genotyped SNP-based heritability of, and genetic correlation between, baseline obesity trait and obesity-change phenotypes.** Left column: heritability ($h_G^2$) estimate means and 95% CI, calculated using the LDSC software[67] on a subset of 1 million HapMap3 SNPs[133] for the following traits: baseline BMI and weight, estimated from intercepts of linear mixed-effects models of obesity traits over time (u0), linear slope change in obesity traits over time (u1 adj. u0), adjusted for intercepts, and posterior probability of membership in a high-gain BMI or weight cluster, adjusted for baseline trait value (prob(k1) adj. u0). Right column: Genetic correlation, $r_G$ means and 95% CI between the obesity-change and baseline obesity. In all panels, summary statistics for correlations and heritability are derived from discovery studies with sample sizes for: BMI = 87,908 females and 73,656 males; weight = 96,264 females and 80,144 males. Circles represent BMI, triangles represent weight; points are coloured by analysis strata (pink: female-sepcific, green: male-specific, grey: sex-combined). *P* values display the level of significance of heterogeneity between the female- and male-specific estimates in each panel.

(Supplementary Data 9, all $P < 1 \times 10^{-16}$). Moreover, the genetic correlation between change in BMI and weight is nearly perfect ($r_G$ for slope terms = 0.98, $r_G$ for posterior probability of membership in gain cluster = 0.95, all $P < 1 \times 10^{-16}$), indicating that the genetic architecture highlighted here is robust to the metric of adiposity used to define trajectories.

## Discussion

In this large-scale EHR- and genetics-based study of longitudinal trajectories of obesity traits, we demonstrate that modelling multiple observations across time increases power to identify genome-wide signals for baseline BMI and weight and enables the discovery of genetic variants associated with changes in adiposity, which are less heritable than and only partially shared with baseline adiposity. Modelling ~1.5 million observations of BMI and weight from >170,000 individuals in the UKBB, enabled us to identify 14 novel, biologically plausible, genetic signals associated with obesity traits. The discovery of these novel loci highlights that repeat measurements can contribute to narrowing the "missing heritability" gap. Leveraging the bespoke longitudinal adiposity phenotypes developed here, we find six genetic loci associated with changes in BMI and weight over time, including a missense variant in *APOE* that replicates in two external cohorts in the United States and Estonia. While previous studies have investigated the associations of cross-sectional BMI SNPs or obesity polygenic scores with adiposity trajectories[15,68], to the best of our knowledge, this study

reports the first genome-wide scan of variants associated with obesity trait trajectories over adulthood.

Accounting for the influence of genetic variation on adiposity change may provide opportunities to personalise obesity prevention and treatment[69,70]. While several studies have investigated the association between BMI-related genetic variants and weight loss guided by medical[70], surgical[71,72], dietary[73], or behavioural[70,74–76] interventions, results are inconsistent across studies, intervention types, and genes assessed. Given our evidence that the genetic basis of adiposity change is distinct from baseline levels, we hypothesise that genetic variants associated with longitudinal weight trajectories may be better predictors of long-term weight change following treatment or lifestyle interventions than variants associated with baseline BMI. Moreover, incorporating information on the genetic signals associated with adiposity trajectories will complement current genetics-based strategies to identify genes for pharmaceutical targets[77] for obesity treatment.

Previous studies have estimated continuity in the genetic correlation of BMI measured at different ages[78], which is theorised to emerge by two possible mechanisms[79]: (1) common genetic (or environmental) factors are associated with the rates of change in BMI over time, which we test in this study, and (2) that these correlations are induced by time-specific genetic (or environmental) factors in an autoregressive manner, i.e., BMI genetics at time-point *t*−1 causally affect BMI at time *t*. Studies testing the latter hypothesis have arrived at

opposing conclusions: Gillespie et al.[80] find that on a genome-wide scale, age-specific genetic effects in an autoregressive framework do not explain differences in BMI heritability across ages 40–73 years, while Winkler et al.[79] did identify 15 genetic loci with differential effects on BMI in younger adults (age <50 years) and older adults (age >50 years). Both studies were pseudo-longitudinal, i.e., the same individuals were not monitored over a period of time, but rather cross-sectional individual data was grouped into age bins. Our work tests a distinct hypothesis and is also, to our knowledge, the first to perform a truly longitudinal genetic study with repeated measures in this age group.

Leveraging EHR to derive longitudinal metrics for genetic discovery may be affected by various biases described earlier[81]. We attempted to mitigate these biases in three ways: (1) While EHR data over-represent sick patients and individuals with higher BMI, UKBB participants are, on average, healthier and have lower BMIs than the population of the UK[82]. Therefore, our UKBB-linked EHR discovery cohort is more overweight than a random sampling of UKBB, but in contrast, UKBB as a whole is ascertained towards lower BMI individuals than a random sampling of the UK. (2) Appending the more accurate UKBB assessment center measurements to the EHR data improves overall data quality. (3) Stringent quality control at both the population and individual increases the signal-to-noise ratio by filtering out a subset of inaccurate data entries. Although we were powered to replicate four of the six UKBB-identified variants for adiposity-change in the MVP cohort, only one replicated; the lack of signal for other variants may imply these are false positive results. However, it is also important to consider the differences in the demographic and obesity-related characteristics between these cohorts, as participants in the MVP are much more likely to have cardiovascular disease and be overweight[44] compared to those in UKBB; and assigning individuals in the former cohort to adiposity trajectory clusters from the latter may distort the phenotypes. Nevertheless, a majority of the baseline adiposity variants in our discovery GWASs as well as the rs429358 variant for adiposity-change replicate across the UKBB, MVP, and EstBB, suggesting that linking EHRs with biobank data may provide a robust framework for genetic discovery.

The two-stage nature of our approach to associate genetic variants with longitudinal trajectories of obesity traits is highly advantageous because of its computational efficiency and convenience. In particular, our method is composable, as the longitudinal analysis of raw data can first be performed separately using a choice of popular, efficient implementations of models; the first-stage outputs can then be taken forward to a GWAS performed in its own bespoke, highly optimised software. The two-stage method approximates the fitting of a full joint model incorporating raw measurement data and genome-wide SNP data. While a full joint model would propagate posterior uncertainty from the longitudinal sub-model through to the GWAS, the approximation here takes forward a single point estimate, i.e. a best linear unbiased predictor (BLUP) or posterior probability of cluster membership, to GWAS. However, in EHR datasets, the number of measurements, and hence estimation precision, can vary across individuals. The propagation of uncertainty between model components, in a similar vein to Markov melding[83], has the potential to further improve the quality of genetic discovery. An interesting area for future research will be to allow for the principled propagation of posterior uncertainty in traits through the highly optimised, multi-locus, mixed-model GWAS methods to perform genetic association in the presence of relatedness and population stratification[84].

It is also important that the choice of trajectory metric utilised in genetic analysis is phenotype-aware. While the variance within an individual's trait value over time may capture meaningful biology for biomarkers such as blood pressure or triglycerides, whose fluctuations are associated with disease development and progress[85,86], weight is a more stable trait that shows a steady pattern of change over many

years[87,88]. Our adiposity-change metrics, derived from regression models incorporating linear and non-linear temporal trends, are better suited to identify the genetic component of BMI and weight trajectories, and are robust to the manner in which this is defined. For example, despite self-report being an imprecise metric[89], lead SNPs from our obesity-change GWASs are also associated with self-reported weight change. However, our results indicate the relative difficulty of identifying genetic associations with longitudinal changes in obesity traits, compared with identifying loci associated with cross-sectional BMI. Variants associated with cross-sectional BMI must have had a causal impact on expected longitudinal BMI at some periods in individuals' lifespans; i.e. a cross-sectional BMI phenotype captures the cumulative longitudinal effects of each BMI-associated genotype up to the age at which the individual is measured. In contrast, our derived measures of longitudinal change target the rate of change of BMI over a shorter average time period, and the magnitude of the genetic signal thus tends to be smaller in the longitudinal analysis compared to the cross-sectional one. This means that the weaker longitudinal genetic signal can be obscured by the non-genetic contribution from individuals' short-and long-term environment, whilst the stronger cross-sectional genetic signal may be detected with higher power as the signal-to-noise ratio is larger. More broadly, there are several factors that might affect the relative power to detect longitudinal effects such as sample size, typically being smaller in longitudinal studies; the longer and more frequent the typical follow-up is in a longitudinal study, the greater the power, and the particular statistical methods used to estimate cross-sectional versus longitudinal traits can affect the accuracy and precision of estimates, and hence the strength of genetic signal detected.

The SNP rs429358 (missense variant in *APOE*) is robustly associated with loss in BMI and weight, independent of baseline obesity, across men and women, across three global cohorts of European ancestry. *APOE* codes for apolipoprotein E, which is a core component of plasma lipoproteins that is essential for cholesterol transport and homoeostasis in several tissues across the body, including the central nervous system, muscle, heart, liver, and adipose tissue[90,91]. The precise pathway by which this variant affects weight change is difficult to pinpoint, as *APOE* is a highly pleiotropic locus associated with hundreds of biomarkers and diseases[63]. Here as well, we find associations between rs429358 and 11 biomarker trajectories. Obesity is cross-sectionally associated with several of these, including levels of triglycerides and cholesterol[92,93], markers of chronic inflammation[94], and haematological traits[95]. Some of the effects of rs429358 are discordant with previously reported phenotypic correlations between obesity and these biomarkers, however, the causal longitudinal and pleiotropic nature of these associations remain to be established. As rs429358 is also the strongest genetic risk factor for Alzheimer's disease[59,60], which is preceded by weight loss[96], we ensured that our findings were robust to the exclusion of individuals with dementia. As longevity may confound the *APOE*-weight loss association[61,62], we adjusted analyses for the length of follow-up in EHR to mitigate against survivor bias; however, we also present age-unadjusted analyses and demonstrate that other lifespan-associated variants are not associated with adiposity change in our GWASs. We thus hypothesise that the *APOE* effect on weight loss may act through cholesterol- and lipid-metabolism pathways that partly determine response to dietary and environmental factors, as seen in mouse models[97,98]. Indeed, it has recently been suggested that *APOE*-mediated cholesterol dysregulation in the brain may influence the onset and severity of Alzheimer's disease[99], suggesting that ageing-associated systemic aberrations in cholesterol homoeostasis could have far-ranging consequences, from weight loss to cognitive decline.

Patterns of weight change in mid-to-late adulthood have been observed to be sex-specific, particularly as women undergo significant changes in weight and body fat distribution around menopause[100].

Here, we find that the heritability of changes in obesity traits is higher in women than in men, supporting a previous finding that obesity polygenic scores are more strongly associated with weight change trajectories in women than in men[68]. This is in contrast to baseline obesity, which is equally heritable in men and women, both in our study and as previously reported[46]. The lower genetic correlation between baseline obesity and obesity-change in women as compared to men, while not statistically significant, may nevertheless indicate sex-differential genome-wide contributions to these phenotypes. We hypothesise that sex hormones could explain some of this sex-specificity, particularly through their role in altering overall obesity and fat distribution around menopause[101,102]. We were underpowered to study the genome-wide architecture of change in adult WC and WHR (10-fold fewer observations than BMI and weight), whose cross-sectional levels are genetically sex-specific with higher heritability in women[46], so more work is needed to disentangle the genetic contribution to changes in adult body fat distribution over time.

While the EHR-linked UKBB cohort has driven genetic discovery for a vast array of human traits in populations of European ancestry[103], sample sizes remain under-powered to detect genome-wide associations in other ancestral groups. We were thus limited to replicating European-ancestry associations in other populations, without the ability to discover ancestry-specific variants associated with adult adiposity trajectories. Furthermore, despite the inclusion of >200,000 individuals in the UKBB EHR data, sample sizes remain low to analyse the genetics of longitudinal trajectory metrics, which have lower heritability than the averaged trait value[15,104] (~7–9x lower in our study) and are thus more challenging to characterise genetically without corresponding increases in sample size. Another limitation of our study was the exclusion of time-varying covariates, such as medication use, smoking status, and other dietary and environmental covariates from models of adiposity change. It is challenging to extract time-dependent values of these variables from EHRs and difficult to ascertain the direction of causality by which these covariates may be associated with weight change. For example, the use of statins to lower blood pressure may be connected to weight gain, mediated indirectly by change in appetite[105], but high blood pressure may itself be a consequence of weight gain[106]. Inappropriate adjustments along this causal pathway may lead to unexpected collider biases[107]. In general, despite their longitudinal nature, it is challenging to assign causality to the associations between weight change and covariates or disease diagnoses from EHR observations alone, as there is no prospective study design to follow[108]. Advances in emulating randomised control trials from longitudinal EHR are beginning to overcome these challenges[109,110], and in the future, it will be critical to incorporate information on genetic risk into these simulated studies.

To the best of our knowledge, this is the largest study to date that characterises the genome-wide architecture of adult adiposity trajectories, and the first to identify specific variants that alter BMI and weight in mid- to late-adulthood. We add evidence to support the growing utility of EHRs in genetics research, and particularly highlight opportunities for incorporating longitudinal information to boost power and identify novel associations. In particular, the *APOE*-associated weight loss identified here contributes to a growing body of evidence on the ageing-associated effects of cholesterol dysregulation. Heterogeneity between men and women in the genome-wide architecture of obesity-change and genetic correlation with baseline obesity highlights the importance of distinguishing between the genetic contributions to mean and lifetime trajectories of phenotypes in sex-specific analyses. In the future, the growing integration of EHR with genetic data in large biobanks will allow us to assess the time-varying associations of rare variants with outsize effects on quantitative traits, as well as to establish genetic and phenotypic relationships among the trajectories of multiple correlated biomarkers across adulthood.

## Methods

### Identification and quality control of longitudinal obesity records

**UK Biobank.** This study was conducted using the UKBB resource, which is a prospective UK-based cohort study with approximately 500,000 participants aged 40–69 years at recruitment, on whom a range of medical, environmental, and genetic information has been collected[42]. Here, we included 409,595 individuals in the white–British ancestry subset identified by Bycroft et al.[111] who passed genotype quality control (QC) (see below).

**Repeat obesity trait measurements.** Obesity-associated traits including BMI and weight were recorded at initial baseline assessment (between 2006 and 2010), as well as at repeat assessments of 20,345 participants (between 2012 and 2013), and at imaging assessments of 52,596 participants (in 2014 and later). We curated a longitudinal research resource by integrating these repeat UKBB assessment centre measurements with the interim release of primary care records provided by GPs for approximately 45% of the UKBB cohort (~230,000 participants, randomly selected)[112] (Supplementary Fig. 3). Each individual with at least one BMI record (coded as Clinical Practice Research Datalink (CPRD) code 22K.) or weight record (coded as CPRD code 22A) in the GP data had their respective UKBB assessment centre measurements appended. Following phenotype and genotype QC, this resulted in 162,666 participants of white–British ancestry with multiple BMI measurements and 177,472 participants with multiple weight measurements (Supplementary Fig. 3).

**Quality control.** We performed both population-level and individual-level longitudinal QC. Participants with codes for history of bariatric surgery (Supplementary Data 10, as identified by Kuan et al.[113]) were excluded entirely, while BMI and weight observations up to the date of surgery were retained for individuals where this could be determined. Only those measures recorded in adulthood (ages 20–80 years) were retained. We excluded implausible observations, defined as more extreme than ±10% of the UKBB assessment centre minimum and maximum values, respectively (BMI <10.9 kg/m² or >82.1 kg/m² and weight <27 kg or >217 kg). We further removed any extreme values >5 SDs away from the population mean to exclude possible technical errors. At the individual-level we excluded multiple observations on the same day, which are likely to be recording errors, by only retaining the observation closest to the individual's median value of the trait across all time points. Finally, we excluded any extreme measurements on the individual-level. For individual $i$ with $J_i$ data points represented as (measurement, age) pairs $(y_{i,j}, t_{i,j})$ for $j = 1, ..., J_i$ ordered chronologically, i.e., $t_{i,1} < ... < t_{i,J_i}$, a "jump" $P_{i,j}$ for $j = 1,...,J_i - 1$ was defined as:

$$P_{i,j} = \log_2 \frac{|y_{i,j+1} - y_{i,j}|/y_{i,j}}{t_{i,j+1} - t_{i,j}} \tag{1}$$

We removed data points associated with extreme jumps (>3 SDs away from the population mean jump, to exclude possible technical errors) by excluding the observation farther from the individual's median value of the trait across all time points.

**BMI and weight validation data.** Participants with BMI and weight observations in UKBB assessment centre measurements who were not included in the interim release of the GP data were held out of discovery analyses (Supplementary Fig. 3). This resulted in 245,447 individuals with at least one BMI observation and 230,861 individuals with at least one weight observation for replication of cross-sectional results. For the replication of longitudinal results, a subset of individuals was used comprising 17,006 individuals with multiple observations of BMI, and 17,035 individuals with multiple observations of weight, from repeat assessment centre visits.

**Self-reported weight change data.** At each UKBB assessment centre visit, participants were asked the question: "Compared with one year ago, has your weight changed?", reported as "No−weigh about the same", "Yes−gained weight", "Yes−lost weight", "Do not know", or "Prefer not to answer". We coded the 1-yr self-reported weight change response at the first assessment centre visit as an ordinal categorical variable with three levels: "loss", "no change", and "gain", excluding individuals who did not respond or responded with "Do not know" or "Prefer not to answer". We retained 301,943 individuals of white−British ancestry who were not included in any of the discovery analyses.

**Abdominal adiposity data.** Similar to the BMI and weight validation datasets, we retained the 44,154 participants with multiple WC and hip circumference (HC) records across repeat assessment centre visits who were not included in the interim release of the GP data, and hence held out of discovery analyses. WHR was calculated at each visit by taking the ratio of WC to HC. We further calculated WC adjusted for BMI (WCadjBMI) and WHR adjusted for BMI (WHRadjBMI) values at each visit for which WC, HC, and BMI were recorded simultaneously by taking the residual of WC and WHR in linear regression models with BMI as the sole predictor.

**Models to define baseline adiposity and adiposity change traits**
Individual $i$ has $J_i$ data points represented as (measurement, age) pairs $(y_{i,j}, t_{i,j})$ for $j = 1, …, J_i$ ordered chronologically, i.e. $t_{i,1} < … < t_{i,J_i}$. The following models are all fitted separately in three strata: female-specific, male-specific, and sex-combined.

**Intercept and slope traits for GWAS.** We implement a two-stage algorithm to estimate and preprocess local intercept and slopes of obesity traits to be taken forward to GWAS in both discovery and validation datasets.

1. Fit random-slope, random-intercept mixed model with the maximum likelihood estimation procedure in the lme4[114] package in R[115]. We target two quantities: the baseline value of each individual's clinical trait (the $\beta_0 + u_{i,0}$ below); and the the linearly approximated rate of change in the trait during each individual's measurement window (the $\beta_1 + u_{i,1}$ below):

$$
\begin{aligned}
y_{i,j} &= x_i^T \gamma + (\beta_0 + u_{i,0}) + (\beta_1 + u_{i,1}) \cdot (t_{i,j} - t_{i,1}) + \varepsilon_{i,j} \\
u_{i,k} &\sim \mathrm{N}(0, \sigma_{u,k}^2), \quad k = 0,1 \\
\varepsilon_{i,j} &\sim \mathrm{N}(0, \sigma_\varepsilon^2),
\end{aligned}
\tag{2}
$$

where individual-specific covariates $x_i$ comprise: baseline age, (baseline age)², data provider, year of birth, and sex. Variance parameters $\sigma_{u,k}^2$ and $\sigma_\varepsilon^2$ are estimated. Fitting model (2) outputs fixed effect model estimates $\hat\gamma$, $\hat\beta_0$, $\hat\beta_1$ and BLUPs of the random effects $\hat u_{i,0}$ and $\hat u_{i,1}$.

2. Linearly adjust and transform the outputted BLUPs. We fit and subtract the linear predictor in each of the linear models:

$$
\hat u_{i,0} = x_{i,0}^T \gamma_0 + \varepsilon_{,0}
\tag{3}
$$

$$
\hat u_{i,1} = x_{i,1}^T \gamma_1 + \varepsilon_{,1}
\tag{4}
$$

where the vector of intercept-adjusting covariates $x_{i,0}$ in (3) comprise: baseline age, (baseline age)², sex, year of birth, assessment centre, number of follow-ups, and total length of follow-up (in years). The vector of slope-adjusting covariates $x_{i,1}$ in (4) comprise the same as $x_{i,0}$ but additionally include the intercept BLUP $\hat u_{i,0}$. The coefficient vectors $\gamma_0$ and $\gamma_1$ in (3) and (4) are estimated by least squares and are distinct from the previously estimated $\gamma$ in (2). We finally apply a deterministic rank-based inverse-normal transformation[116] to the

residuals from fitting models (3) and (4). For example, the intercept trait for individual $i$ taken forward to GWAS is

$$
\tilde u_{i,0} = \Phi^{-1}\left( \frac{r(\hat u_{i,0} - x_{i,0}^T \hat\gamma_0) - c}{N - 2c + 1} \right)
\tag{5}
$$

where $r(\hat u_{i,0} - x_{i,0}^T \hat\gamma_0)$ is the rank of the $i$th residual among all $N$ residuals, the offset $c$ is 0.5, and $\Phi(\cdot)$ is the cumulative distribution function (CDF) of the standard Gaussian distribution.

The distribution of residuals and BLUPs from the LME models are heavy-tailed relative to a Gaussian (Supplementary Figs. 10–12). Such model misspecification could potentially lead to miscalibration of CIs and hypothesis tests based on the standard linear mixed model, although this is likely to be mitigated by the large sample size owing to the central limit theorem. We therefore take forward covariate-adjusted and inverse-normal transformed BLUPs, as described in (5), for genome-wide association testing.

**Modelling non-linear trajectories with regularised splines.** We model non-linear changes in obesity traits using a regularised B-spline basis of degree 3 (i.e., a cubic spline model) with $n_{\mathrm{df}} = 100$ degrees of freedom, incorporating $n_{\mathrm{df}} - 4$ (i.e., $n_{\mathrm{df}} - 3[\mathrm{degree}] - 1$ [intercept]) knots that are spaced evenly across each individual's first $T = 7500$ post-baseline days $\approx 20.5$ years. It is common practice in semi-parametric regression to use regularised splines with a relatively large number of knots, thereby allowing functional expressiveness without overfitting[31,117]. Conditional on the spline coefficients, $b_i$, the likelihood for measurements $y_i$ (individual $i$'s $J_i$-vector of measurements taken at days $t_{i,1}, …, t_{i,J_i}$) is

$$
p(y_i | b_i, \sigma^2) = \mathrm{MVN}(y_i | Z_i X_B b_i, I \sigma^2)
\tag{6}
$$

where: the $n_{\mathrm{df}}$-vector $b_i$ contains the $i$th individual's spline basis coefficients; $X_B$ is the $(T+1) \times n_{\mathrm{df}}$ matrix of spline basis functions evaluated at days 0, …, $T$ post-baseline; and $Z_i$ is a $J_i \times (T+1)$ matrix whose $j$th row extracts day $t_{i,j} - t_{i,1}$ post-baseline, i.e.,

$$
[Z_i]_{j,k} = \begin{cases} 1 & \text{if } k = t_{i,j} - t_{i,1} + 1 \\ 0 & \text{otherwise} . \end{cases}
$$

We specify an order-1 autoregressive ($AR(1)$) model as a smoothing prior on spline coefficients, $b_i$, which vary smoothly around an individual-specific mean value, $\mu_i$. On $\mu_i$ we specify a non-informative prior: $\mathrm{N}(\mu_i | 0, \sigma_\mu^2)$ with large SD $\sigma_\mu$. The resulting $\mu_i$-marginalised prior for $b_i$ is

$$
\begin{aligned}
p(b_i) &= \mathrm{MVN}(b_i | 0, \Sigma_B) \\
\Sigma_B &:= \Sigma_{AR(1)} + \sigma_\mu^2 \vec{\mathbf{1}} \\
\left[\Sigma_{AR(1)}\right]_{k,k'} &:= \sigma_{AR(1)}^2 \phi^{|k-k'|},
\end{aligned}
\tag{7}
$$

where: $\Sigma_{AR(1)}$ is the $n_{\mathrm{df}} \times n_{\mathrm{df}}$ autocovariance matrix implied by an $AR(1)$ model with lag-1 autocorrelation $\phi \in [0,1)$ and scale parameter $\sigma_{AR(1)}^2 > 0$; and $\vec{\mathbf{1}}$ is an $n_{\mathrm{df}} \times n_{\mathrm{df}}$ matrix of ones.

The prior at (7) and likelihood at (6) are a specific case of the Bayes linear model[118], for which the posterior is available in closed form:

$$
\begin{aligned}
p(b_i | y_i, \Sigma_B, \sigma^2) &= \mathrm{MVN}(b_i | m_i, \sigma^2 V_i) \\
V_i &:= \left( X_B^T Z_i^T Z_i X_B + \Sigma_B^{-1} \right)^{-1} \\
m_i &:= V_i X_B^T Z_i^T y_i.
\end{aligned}
\tag{8}
$$

The posterior at (8) can be evaluated separately and in parallel across individuals because the $(y_i, b_i)$ are conditionally independent

across individuals $i$ given the hyperparameters $\sigma^2_{AR(1)}$, $\phi$, $\sigma_\mu$ and $\sigma^2$. Values of hyperparameters in the smoothing prior are chosen subjectively, via visualisation of randomly selected samples of individual data trajectories, to reflect empirical levels of smoothness: $\sigma^2_{AR(1)} := 2.5$, $\phi := 0.99$, $\sigma_\mu := 100$ (Supplementary Fig. 4). We additionally compared cluster allocations for 5000 randomly selected individuals across the following settings of hyperparameters: ($\sigma^2_{AR(1)} := 0.5$, $\phi := 0.9$, $\sigma_\mu := 10$), ($\sigma^2_{AR(1)} := 2.5$, $\phi := 0.99$, $\sigma_\mu := 100$), and ($\sigma^2_{AR(1)} := 10$, $\phi := 0.999$, $\sigma_\mu := 500$) (Supplementary Fig. 8).

For each trait separately, we set $\sigma^2$ to the median of its individual-specific maximum likelihood estimates (MLEs), i.e., $\sigma^2 :=$ median $\{\frac{1}{J_i}||\mathbf{y}_i - \mathbf{Z}_i\mathbf{X}_B\mathbf{m}_i||^2_2 : i = 1, \dots, n\}$ where each MLE is calculated from (6) after substituting for $\mathbf{b}_i$ its maximum a posteriori estimate, $\mathbf{m}_i$ from (8) (Supplementary Data 12).

The measurements $\mathbf{y}_i$ inputted into the likelihood for the regularised spline model at (6) are pre-processed by taking the standardised residual from the linear model with the following covariates: baseline age, (baseline age)$^2$, data provider, year of birth, and sex, i.e. from the model $y_{i,j} = x_i^T\gamma + \varepsilon_{i,j}$ fitted across all $i = 1, \dots, N$ individuals and $j = 1, \dots, J_i$ time points. Standardisation of residuals then proceeds by subtracting the mean and dividing by the SD of residuals across all individuals and time points.

We focus on individual $i$'s posterior change from baseline, i.e. on

$$\tilde{\mathbf{b}}_i := (0, u_{i,2} - u_{i,1}, u_{i,3} - u_{i,1}, \dots)^T \tag{9}$$

$$\equiv \mathbf{Db} \tag{10}$$

where the $j$th row of $\mathbf{D}$ is $(\mathbf{e}_j - \mathbf{e}_1)^T$ and $\mathbf{e}_k$ is the $k$th basis vector, i.e. a column $n_{df}$-vector with zeroes everywhere except the $k$th entry, which is one. To calculate the posterior for $\tilde{\mathbf{b}}_i$ we linearly transform the posterior at (8) so that

$$p(\tilde{\mathbf{b}}_i|\mathbf{y}_i, \mathbf{\Sigma}_B, \sigma^2) = \text{MVN}(\tilde{\mathbf{b}}_i|\mathbf{Dm}_i, \sigma^2\mathbf{DV}_i\mathbf{D}^T) \tag{11}$$

with $\mathbf{m}_i$ and $\mathbf{V}_i$ defined at at (8).

**Soft clustering of individuals by non-linear adiposity trajectory patterns.** See Supplementary Fig. 5 for an overview of the clustering protocol.

Any two individuals typically have quite distinct measurement profiles, with different numbers of measurements taken at ages which may be quite disparate. Therefore the precision with which we can estimate any particular spline coefficient varies across individuals. To incorporate this heteroscedasticity into our clustering framework, we define the following scaled Euclidean distance between each pair of individuals $(i, i')$ in the space of baselined spline basis coefficients:

$$d(i, i') = \sqrt{\sum_{k=1}^{n_{df}} \frac{([\mathbf{Dm}_i]_k - [\mathbf{Dm}_{i'}]_k)^2}{([\mathbf{DV}_i\mathbf{D}^T]_{k,k} + [\mathbf{DV}_{i'}\mathbf{D}^T]_{k,k})\sigma^2}} \tag{12}$$

where $\mathbf{m}_i$ and $\sigma^2\mathbf{V}_i$ are the posterior mean and covariance of individual $i$'s spine coefficients $\mathbf{b}_i$ taken from (8). For each spline coefficient $k$ in (12), the squared difference between individuals' $i$ and $i'$ mean coefficients is standardised by the sum of the corresponding variances.

We perform $k$-medoids clustering using the partitioning around medoids (PAM) algorithm[55,56] as implemented in the `pam` function in the `cluster` package[119] in R[115]. We train cluster centroids on a randomly selected subset of 80% of individuals in each analysis strata. We filter individuals in the training set to retain only those with at least $L = 2$ observations. For a fixed number of clusters, $K = 4$, we initialise cluster membership according to bins $\mathcal{B}_{1:K}$ demarcated by the $0, \frac{1}{K}, \frac{2}{K}, \dots, 1$ empirical quantiles of the estimated fold change in obesity

trait between baseline and year $M = 2$:

$$\mathcal{B}_k := \left[\hat{F}^{-1}\left(\frac{k-1}{K}\right), \hat{F}^{-1}\left(\frac{k}{K}\right)\right) k = 1, \dots, K$$

$$\hat{F}(\cdot) := \text{empirical CDF of } \left\{\frac{[\mathbf{X}_B\mathbf{Dm}_i]_{M+1}}{[\mathbf{X}_B\mathbf{Dm}_i]_1} : i = 1, \dots, N\right\}$$

individual $i$ in bin $k \iff \frac{[\mathbf{X}_B\mathbf{Dm}_i]_{M+1}}{[\mathbf{X}_B\mathbf{Dm}_i]_1} \in \mathcal{B}_k.$

(13)

To ensure robustness, we run the clustering algorithm $S = 10$ times, each on a random sub-sample of size 5000 (without replacement). For each clustering output $s = 1, \dots, S$, we calculate the point-wise mean of each cluster's constituent individuals:

$$\mathbf{c}_{k,s} := \frac{1}{|\mathcal{C}_k^{(s)}|} \sum_{i \in \mathcal{C}_k^{(s)}} \mathbf{Dm}_i \tag{14}$$

For each clustering $s$, we observe all trajectories $\mathbf{c}_{s,1:K}$ to be monotonic and non-overlapping (Supplementary Fig. 6). We can therefore define ordered cluster means $\mathbf{c}_{(k),s}$,

$$k < k' \iff [\mathbf{c}_{(k),s}]_j > [\mathbf{c}_{(k'),s}]_j \quad \forall j = 1, \dots, n_{df}, \tag{15}$$

and average the $k$th ordered mean across $S$ clusterings, where the highest-weight cluster mean is given by $\mathbf{c}_{(1)}$ and the lowest by $\mathbf{c}_{(K)}$:

$$\mathbf{c}_{(k)} := \frac{1}{S}\sum_{s=1}^{S} \mathbf{c}_{(k),s}, \tag{16}$$

with corresponding point-wise SEs. We investigate the sensitivity of the resulting clusters to number of clusters $K$, filter parameter $L$ (minimum number of measurements), and the cluster initialisation parameter $M$ appearing in (13) via silhouette values[120], which evaluate the similarity between cluster members (cohesion) vs others (separation) (Supplementary Fig. 6). We test values of $K$ from 2, ..., 8, filtering parameter $L \in (2, 5, 10)$, and initialisation parameter $M \in (1, 2, 5, 10)$ or random initialisation to choose a combination of parameters that produces dense and separable clusters, i.e. $K = 4$, $L = 2$, $M = 2$. We also qualitatively evaluate cluster centroids across all parameter settings (Supplementary Fig. 7). Finally, we compared cluster allocations over each of the 10 random trains for a set of 5000 randomly sampled individuals held out of the training splits (Supplementary Fig. 9).

Once cluster centroids have been calculated, we define individual $i$'s soft cluster membership probability of belonging to cluster $k$ as the posterior probability of being closest in Euclidean distance to cluster $k$'s centroid:

$$\pi_{i,(k)} := \int \mathbb{I}\left(k = \underset{k'}{\text{argmin}} ||\tilde{\mathbf{b}}_i - \mathbf{c}_{(k')}||_2\right) \text{MVN}(\tilde{\mathbf{b}}_i|\mathbf{Dm}_i, \sigma^2\mathbf{DV}_i\mathbf{D}^T)d\tilde{\mathbf{b}}_i \tag{17}$$

where the second term in the integrand is the posterior from (8), and we approximate the integral in (17) using 100 Monte Carlo samples from the posterior.

Finally, we validate the clustering by comparing cluster properties of the randomly selected 80% training set used to define cluster centroids, with the held-out 20% validation set. We assign each individual to the cluster for which they have highest membership probability and compare the proportion of individuals assigned to each cluster, as well as distributions of sex, baseline age, number of follow-up measures, and total length of follow-up of individuals assigned to each cluster. These metrics are similar across training and validation sets in all strata (Supplementary Data 13).

Finally, we take forward bounded logit-transformed cumulative cluster probabilities to GWAS. These outputs are defined as bounded $\text{logit}(\pi_{i,(1)})$, bounded $\text{logit}(\pi_{i,(1)} + \pi_{i,(2)})$, and bounded $\text{logit}(\pi_{i,(1)} + \pi_{i,(2)} + \pi_{i,(3)})$, i.e., the bounded log odds of being in the highest (k1), highest two (k1 or k2), and highest three (k1, k2, or k3) weight clusters respectively. To prevent infinite log odds at $\pi \in \{0, 1\}$ we defined the following bounded logit transform[121]:

$$\text{bounded logit}(\pi) \equiv \text{logit}\left(\frac{(S-1)\pi + 0.5}{S}\right) \quad \pi \in [0,1], \quad (18)$$

where $S = 100$, the number of Monte Carlo samples from the posterior in approximating (17).

## Genome-wide association studies
### QC of UK Biobank genotyped and imputed data.
Genotyping, initial genotype QC, and imputation on genome build hg19 were performed by UKBB[111]. We performed post-imputation QC to retain only bi-allelic SNPs with MAF >0.01, info score >0.8, missing call rate < 5%, and Hardy-Weinberg equilibrium (HWE) exact test $P > 1 \times 10^{-6}$. We additionally performed sample QC to exclude individuals with sex chromosome aneuploidies, whose self-reported sex did not match inferred genetic sex, with an excess of third-degree relatives in UKBB, identified as heterozygosity or missingness outliers, excluded from autosome phasing or kinship inference, and any other UKBB recommended exclusions[111].

### Linear mixed model association analyses for quantitative traits.
An overview of the traits carried forward for GWAS is provided in Supplementary Fig. 18. The following association analyses are all performed separately in three strata: female-specific, male-specific, and sex-combined. The intercept and slope traits for GWAS, i.e., $\tilde{u}_{i,0}$ and $\tilde{u}_{i,1}$ were tested for association with genetic variants, adjusted for the first 21 genetic principal components (PCs) and genotyping array, using the BOLT-LMM software[84]. We also performed GWAS for the inverse-normal transformed within-individual mean adiposity trait, adjusting for the same covariates described for $\tilde{u}_{i,0}$. A similar protocol was followed for the logit-transformed soft clustering probability traits, i.e. $\pi''_{i,1}$, $\pi''_{i,2}$, and $\pi''_{i,3}$ with additional adjustments for baseline trait, baseline age, (baseline age)$^2$, sex, year of birth, assessment centre, number of follow-ups, and total length of follow-up (in years).

### Fine-mapping SNP associations.
We identified putative causal variants at all GWS loci (defined by merging windows of 1.5 Mb around SNPs with $P < 5 \times 10^{-8}$), using FINEMAP[122] to select variants (lead SNPs) with a posterior inclusion probability >95%. Lead SNPs were annotated to the nearest gene transcription start site.

### Classifying baseline BMI and weight SNPs as reported, refined, or novel obesity associations.
We curated a list of SNPs associated with any of 44 obesity-related traits in the GWAS Catalog[54] accessed on 02 Nov 2021, henceforth referred to as published obesity-associated variants (Supplementary Data 1). We then conducted conditional analysis using GCTA-COJO[123] for each lead SNP in our GWAS and published obesity-associated variants within 500 kb, classifying variants as reported, refined, or novel based on previously recommended criteria[47]. Reported SNPs in our study are those whose effects are fully accounted for by published obesity-associated variants within 500 kb. Refined SNPs fulfil all of the following criteria: (1) the refined SNP is correlated (linkage disequilibrium (LD) $r^2 \geq 0.1$) with at least one published obesity-associated variant within 500 kb, (2) the refined SNP has a significantly stronger effect ($P < 0.05$ in a two-sample $t$ test for difference in mean effect sizes) on the BMI- or weight-intercept trait than published obesity-associated SNPs and also accounts for the effect of published obesity-associated SNPs in conditional analysis (conditional

$P > 0.05$), and (3) published obesity-associated SNPs cannot fully account for the effect of the refined SNP in conditional analysis (conditional $P < 0.05$). Finally, a SNP in our study was declared novel if it was not in LD with ($r^2 < 0.1$), and conditionally independent of (conditional $P < 0.05$), all published obesity-associated variants within 500 kb.

### Replication of GWS associations in UK Biobank hold-out sets
**BMI and weight intercept-trait genetic associations.** We created cross-sectional obesity phenotypes for the 245,447 individuals in the hold-out set for BMI and 230,861 individuals in the hold-out set for weight (Supplementary Fig. 3) by retaining the observed trait value closest to the individual's median trait value (if multiple observations present). Deterministic rank-based inverse-normal transformation[116] was applied to the residual of the obesity trait adjusted for age, age$^2$, year of birth, data provider, and sex. We then tested this trait for association with genetic variants, adjusted for the first 21 genetic PCs and genotyping array, using the BOLT-LMM software[84].

**BMI and weight slope-trait genetic associations.** We created adiposity slope phenotypes for the 17,006 individuals with multiple observations of BMI and 17,035 individuals with multiple observations of weight from repeat assessment centre visits (Supplementary Fig. 3 and Supplementary Data 19) with BLUPs from LMEs models as described in the slope-trait modelling section above. We tested for association of this slope-trait with GWS variants associated with adiposity change in our discovery analyses, adjusted for the first 21 genetic PCs and genotyping array, via the linear regression framework implemented in PLINK[124]. As PLINK does not account for family structure, we compared each pair of second-degree or closer related individuals (kinship coefficient >0.0884)[111] and excluded the individual in the pair having higher genotyping missingness. We repeated the same protocol within each self-identified ethnic group of individuals not of white–British ancestry (Supplementary Data 11).

**Genetic associations with BMI and weight cluster probabilities.** We fit regularised splines as detailed above to the 17,006 individuals with multiple observations of BMI and 17,035 individuals with multiple observations of weight from repeat assessment centre visits (Supplementary Fig. 3). Soft cluster membership probabilities for these individuals were calculated, and the three logit-transformed $\pi_i$ traits were carried forward for association testing with GWS variants associated with adiposity change in our discovery analyses. As above, we pruned out second-degree or closer related individuals and performed association analysis, adjusted for baseline trait, baseline age, (baseline age)$^2$, assessment centre, first 21 genetic PCs and genotyping array, via the linear regression framework implemented in PLINK[124]. We repeated the same protocol within each self-identified ethnic group of individuals not of white–British ancestry.

**Genetic associations with self-reported weight change.** We fit proportional odds logistic regression models implemented in the MASS package[125] in R[115] to estimate the additive effect of lead SNPs on self-reported one-year weight change coded as an ordinal categorical variable with three levels: "loss", "no change", and "gain" in 301,943 individuals (described in the data section above). All models were adjusted for BMI, age, sex, year of birth, data provider, assessment centre, first 21 genetic PCs and genotyping array. We repeated the same protocol within each self-identified ethnic group of individuals not of white–British ancestry.

### Replication of GWS associations in external cohorts
Quality control, modelling of adiposity change, and GWAS in external cohorts were all performed exactly as in the UKBB discovery analyses, with any exceptions noted below.

**Million Veteran Program.** The MVP mega-biobank, with ~950,000 participants enroled to date, is actively recruiting participants from the 6.9 million eligible individuals who make use of the services provided by the Veterans Health Administration (VHA) from around 50 Veterans Affairs (VA) facilities across the United States of America (USA)[43]. Eligible candidates are registered VHA users who are at least 18 years of age, possess a valid mailing address, and have the ability to provide informed consent. The VA Central Institutional Review Board (IRB) 10-02 protocol gained approval from the VA Central IRB in 2010, and the enrolment of study participants commenced in early 2011. Genetic data for this study was obtained from the custom-genotyped dataset with imputation to the 1000 Genomes project on genome build hg19, and filtered to markers with imputation information score >0.30 with minor allele count >30[126]. Full characteristics of the MVP cohort[43] and associated genetic data[126] have been described previously.

Weight, height, and other covariate records were compiled from the MVP Baseline Survey, which collected information on demographics, health status, lifestyle habits, military experience, and physical traits, and supplemented with EHRs. A survey cleaning algorithm was used to process self-reported data, ensuring quality through expert-defined rules, full details of which have been described previously[44]. Following population-level and individual-level QC of repeat BMI measurements as described above, we retained 404,503 male European-ancestry participants with 20.6 million observations of BMI and 33,200 female European-ancestry participants with 1.94 million observation of BMI.

For each participant, we calculated linear rates of change in BMI over time with the LME models described in (2); we also calculated each individual's soft cluster membership probability of belonging to clusters whose centroids were defined in the UKBB discovery data (Supplementary Data 24). All analyses were performed in sex-specific and sex-combined strata. Genetic association analysis was performed using `REGENIE v2.2.4`, software for whole genome regression modelling of large GWASs that accounts for relatedness and population stratification[127]. All GWASs were adjusted for baseline age, (baseline age)$^2$, the first 10 genetic PCs, and sex (in sex-combined analyses).

**Estonian biobank.** EstBB is a volunteer-based sample of Estonian residents comprising ~20% of the Estonian adult population ($N > 210,000$), recruited by medical personnel and through media campaigns. Various health and demographic data have been collected from the participants, both by medical workers and via self-reports, since 2002. The cohort has been described in detail by Leitsalu et al.[45]. Genetic data for this study was obtained from genotyping with the Illumina global screening array (GSA) microchip, with imputation using a customised reference panel aligned to the hg19 genome, as described previously[128].

BMI was available for 193,490 participants. BMI measurements were collected by doctors (through measurements of height and weight) from 2001 to 2023. Population-level and individual-level QC of repeat BMI measurements were performed as described for the UKBB discovery cohort; we additionally excluded individuals with records of use of GLP-1 inhibitors such as semaglutide (blood glucose-lowering drugs that typically also result in weight loss, drug codes A10BJ*). In total, 82,034 female participants with 281,438 measurements of BMI and 45,735 male participants with 164,166 measurements of BMI were retained. Of these, 125,209 passed genotyping QC.

For each participant, we calculated linear rates of change in BMI over time with the LME model described in (2); we also calculated each individual's soft cluster membership probability of belonging to clusters whose centroids were defined in the UKBB discovery data (Supplementary Data 24). All analyses were performed in sex-specific and sex-combined strata. Genetic association analysis was performed using `REGENIE v3.2` software for whole genome regression modelling[127].

All GWASs were adjusted for baseline age, (baseline age)$^2$, the first 20 genetic PCs, and sex (in sex-combined analyses).

## Power calculations for replication sample sizes
We corrected the observed effect sizes from discovery GWASs for winner's curse through an implementation first described by Palmer et al.[129]. Briefly, we solve for the bias using the following maximum likelihood model,

$$\beta_{obs} = \beta_{true} + s \frac{\phi\left(\frac{\beta_{true}}{s} - c\right) - \phi\left(\frac{-\beta_{true}}{s} - c\right)}{\psi\left(\frac{\beta_{true}}{s} - c\right) + \psi\left(\frac{-\beta_{true}}{s} - c\right)} \tag{19}$$

where $\beta_{obs}$ is the effect size in the discovery GWAS, $\beta_{true}$ is the (assumed true) effect size in the source population, and $c = 5.33$ is the test statistic corresponding to a discovery $\alpha = 5 \times 10^{-8}$. The sample size required to replicate the (assumed true) unbiased effect size is then calculated for nominally significant $\alpha = 0.05$ and Bonferroni-adjusted for the number of independent variants tested, $M_{var}$ ($\alpha = \frac{0.05}{M_{var}}$) as follows:

$$\text{power}(\alpha, ncp) = 1 - \chi_1^2((\chi_1^2)^{-1}(1 - \alpha), ncp) \tag{20}$$

under the alternative distribution which is non-central $\chi_1^2$ with non-centrality parameter per variant (ncp) estimated for a normalised trait with variance 1 as:

$$ncp \approx N \frac{2\beta_{obs}^2 \text{AF}(1 - \text{AF})}{1 - 2\beta_{obs}^2 \text{AF}(1 - \text{AF})} \tag{21}$$

where AF is the variant allele frequency.

## Power comparison to GIANT 2019 meta-analysis of BMI
We accessed publicly available summary statistics from the GIANT consortium's meta-analysis of BMI across UKBB and previous GIANT releases in female-specific (max $N = 434,793$), male-specific (max $N = 374,755$), and sex-combined strata (max $N = 806,834$)[46]. SNPs included in both the GIANT 2019 meta-analysis and our in-house BMI-intercept GWAS that reached GWS in either study were carried forward for power comparisons, resulting in 26,812 (female-specific strata), 22,123 (male-specific strata), and 82,559 (sex-combined strata) SNPs. Per variant, we calculated the $\chi^2$ statistic (as $\frac{\beta^2}{SE^2}$) and obtained the ratio of $\chi_{in-house}^2$ to $\chi_{GIANT}^2$. Median $\frac{\chi_{in-house}^2}{\chi_{GIANT}^2}$ across all GWS SNPs was then compared to the median ratio of sample sizes, i.e. $\frac{N_{in-house}}{N_{GIANT}}$, to determine the boost in power over that expected from the sample size difference between the two studies.

## Single-variant analyses
The following analyses were all conducted in female-specific, male-specific, and sex-combined strata.

**Abdominal adiposity change traits.** Slope changes in WC, WHR, WCadjBMI, and WHRadjBMI for up to 44,154 individuals with repeat observations were calculated using LMEs models, adjusted and rank-based inverse-normal transformed[116] for genetic association testing as described in the slope modelling section above. We estimated the additive association of number of copies of each lead variant minor allele (0, 1, or 2) with slope traits adjusted for the first 21 genetic PCs and genotyping array via linear regression (Supplementary Data 17).

**Longitudinal phenome-wide association.** We curated a longitudinal research resource for 45 additional quantitative phenotypes in up to

146,099 individuals of white–British ancestry (Supplementary Data 14, as identified by Kuan et al.[130]) by integrating UKBB assessment centre measurements with the interim release of primary care records provided by GPs, with QC performed as described above for obesity traits. Slope changes in each of these phenotypes were calculated using LMEs models described in (2). A deterministic rank-based inverse-normal transformation[116], as described in (5), was applied to the slope BLUP $\hat{u}_{i,1}$. The transformed slope-trait was tested for additive association with number of copies of each lead variant minor allele (0, 1, or 2), adjusted for the intercept BLUP $\hat{u}_{i,0}$, baseline age, (baseline age)2, sex, year of birth, number of follow-ups, total length of follow-up (in years), assessment centre, first 21 genetic PCs and genotyping array (Supplementary Data 18).

**Identification of individuals with Alzheimer's or dementia diagnoses.** We identified participants with codes for history or diagnosis of dementia in either primary care or hospital in-patient records (Supplementary Data 15, as identified by Kuan et al.[113]). We performed sensitivity analyses for the replication of rs429358 associations with all obesity-change phenotypes after excluding up to 242 individuals of white–British ancestry with recorded history or diagnosis of dementia.

**Identification of lifespan-associated variants**
We curated a list of 138 independent variants associated with longevity in the GWAS Catalog[54], accessed on 27 March 2023 (Supplementary Data 16). We identified independent SNPs that passed genotyping and imputation QC filters in UKBB by pair-wise pruning variants in LD ($r^2 > 0.1$) within a 1 Mb window. One of the lead variants identified in this study, i.e., rs429358 in the *APOE* locus, was pruned out in favour of rs4420638, which is 11 kb away from the lead variant and in LD with rs429358 with $r^2 = 0.69$. We looked up the effects of these variants in the various adiposity-change GWAS summary statistics and established significance at $P = 3.60 \times 10^{-4}$ (Bonferroni-corrected at 5% across 138 tests).

**SNP heritability and genetic correlations**
We estimated the heritability explained by genotyped SNPs ($h_G^2$) and genetic correlations ($r_G$) between obesity-intercept and obesity-change traits, from summary statistics, using LD score regression implemented in the LDSC software[67,131], with pre-computed LD-scores based on European-ancestry samples of the 1000 Genomes Project[132] restricted to HapMap3 SNPs[133]. The same protocol was followed to determine $r_G$ between BMI-intercept in our in-house study and BMI in the GIANT 2019 meta-analysis.

**Joint modelling of intra-individual mean and variance**
Analyses were performed using the TrajGWAS package[28] in Julia[134], for 177,472 unrelated individuals of white–British ancestry with multiple measurements of weight included in the discovery analyses. Briefly, TrajGWAS analysis is conducted in two stages to test for genetic effects on longitudinal trajectory mean, intra-individual variance, and a joint effect on either mean or variance in an LME model framework[28]. In the first stage, we fit a null model for weight with fixed effects for the intercept, age, age$^2$, sex, and 21 genetic PCs; we included random effects for the intercept and linear slope of age. In the second stage, we performed score testing with the saddle-point approximation under the full model, i.e. including genome-wide effects for all variants with MAF >1% in the genotyped and imputed UKBB data that passed QC.

**Sex-heterogeneity testing**
We tested for sex-heterogeneity in the effects of adiposity-change lead SNPs by calculating Z-statistics and corresponding P-values for the difference in female-specific and male-specific effects as:

$$Z_{sexhet} = \frac{(\hat{\beta}_{(F)} - \hat{\beta}_{(M)})}{\sqrt{(SE_{(F)}^2 + SE_{(M)}^2)}} \tag{22}$$

A similar statistic and test was used to determine heterogeneity between ($h_G^2$) of all traits in males and females, and $r_G$ between obesity-intercepts and obesity-change traits in males and females.

**Reporting summary**
Further information on research design is available in the Nature Portfolio Reporting Summary linked to this article.

## Data availability
The GWAS summary statistics generated in this study have been deposited in the GWAS Catalog[54]. They can be downloaded from the parent directory: ftp://ftp.ebi.ac.uk/pub/databases/gwas/summary_statistics/GCST90429001-GCST90430000/ using the accession numbers provided in Supplementary Data 26 (ranging from GCST90429765 to GCST90429794).

## Code availability
All code required to reproduce analyses is publicly available at: https://github.com/lindgrengroup/longitudinal_primarycare/tree/main/adiposity/scripts/manuscript[135].

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

## Acknowledgements

S.S.V. was supported by the Rhodes Scholarships, Clarendon Fund, and the Medical Sciences Doctoral Training Centre at the University of Oxford. K.C. was supported by the University of Leicester (College of Life Sciences) and Health Data Research UK. K.A. was supported by the Estonian Research Council's Personal Starting Grant PSG759. L.B.L.W. was supported by the Wellcome Trust. U.V. was supported by the Estonian Research Council's Personal Starting Grant PSG759. C.H. wishes to acknowledge support from the Alan Turing Institute, the EPSRC grant Bayes4Health, Novartis, and Novo Nordisk. C.M.L. is supported by the Li Ka Shing Foundation, NIHR Oxford Biomedical Research Centre, Oxford, NIH (1P50HD104224-01), Gates Foundation (INV-024200), and a Wellcome Trust Investigator Award (221782/Z/20/Z). G.N. acknowledges funding from the NIHR Biomedical Research Centre, Oxford (grant no. NIHR203311). This research has been conducted using the UK Biobank Resource under Application Number 11867. This research was partially supported by the Wellcome Trust Core Award Grant Number 203141/Z/16/Z with additional support from the NIHR

Oxford BRC. The views expressed are those of the authors and not necessarily those of the NHS, the NIHR, or the Department of Health. This research is partially supported by funding from the Department of Veterans Affairs Office of Research and Development, Million Veteran Program Grant I01-BX003340 and I01-BX004821. This publication does not represent the views of the Department of Veterans Affairs or the United States Government. This study was partially funded by the European Union through the European Regional Development Fund Project No. 2014-2020.4.01.15-0012 GENTRANSMED. Data analysis was carried out in part in the High-Performance Computing Centre of the University of Tartu. The activities of the EstBB are regulated by the Human Genes Research Act, which was adopted in 2000 specifically for the operations of the EstBB. Individual-level data analysis in the EstBB was carried out under ethical approvals of 1.1-12/1409 and 1.1-12/2161 from the Estonian Committee on Bioethics and Human Research (Estonian Ministry of Social Affairs), using data according to release application 6-7/GI/31993 from the EstBB.

## Author contributions

S.S.V., G.N., and C.M.L. conceptualised the study. Data curation and formal analyses were conducted by S.S.V., Kayesha C., G.V.L., Q.H., K.A., U.V., and G.N. S.S.V., H.G., and G.N. developed methodology and software. Data collection was performed by P.W., Y.H., and Kelly C. Funding was acquired by U.V., Y.V.S, C.H., and C.M.L. C.H., G.N., and C.M.L. were responsible for supervision. S.S.V. and G.N. wrote the original draft. S.S.V, H.G., D.S.P., L.B.L.W., C.N., C.H., C.M.L., and G.N. edited the draft.

## Competing interests

L.B.L.W. is currently employed by Novo Nordisk Research Centre Oxford but, while she conducted the research described in this manuscript, was only affiliated with the University of Oxford. C.H. reports grants from Novo Nordisk and Novartis; C.M.L. reports grants from Bayer AG and Novo Nordisk and has a partner who works at Vertex. The other authors declare no competing interests.

## Additional information

[1]Wellcome Centre for Human Genetics, Nuffield Department of Medicine, University of Oxford, Oxford, UK. [2]Big Data Institute, Li Ka Shing Centre for Health Information and Discovery, University of Oxford, Oxford, UK. [3]Department of Statistics, University of Oxford, Oxford, UK. [4]Nuffield Department of Population Health, Medical Sciences Division, University of Oxford, Oxford, UK. [5]Department of Population Health Sciences, University of Leicester, Leicester, UK. [6]Department of Epidemiology, Emory University Rollins School of Public Health, Atlanta, GA, USA. [7]Atlanta VA Health Care System, Decatur, GA, USA. [8]Department of Medicine, Emory University School of Medicine, Atlanta, GA, USA. [9]Massachusetts Veterans Epidemiology Research and Information Center (MAVERIC), Veterans Affairs Boston Healthcare System, Boston, MA, USA. [10]Division of Aging, Brigham and Women's Hospital, Harvard Medical School, Boston, MA, USA. [11]Institute of Psychology, Faculty of Social Sciences, University of Tartu, Tartu, Estonia. [12]Novo Nordisk Research Centre Oxford, Oxford, UK. [13]Nuffield Department of Women's and Reproductive Health, Medical Sciences Division, University of Oxford, Oxford, UK. [14]Estonian Genome Centre, Institute of Genomics, Faculty of Science and Technology, University of Tartu, Tartu, Estonia. [15]Department of Neurology and Neurosurgery, Faculty of Medicine and Health Sciences, University of McGill, Montreal, Canada. [16]Nuffield Department of Medicine, Medical Sciences Division, University of Oxford, Oxford, UK. [17]The Alan Turing Institute, London, UK. [18]Broad Institute of Harvard and MIT, Cambridge, MA, USA. ✉e-mail: samvida@well.ox.ac.uk; cecilia.lindgren@bdi.ox.ac.uk; george.nicholson@stats.ox.ac.uk

## Million Veteran Program

Gregorio V. Linchangco Jr.[6,7], Qin Hui ®[6,7], Peter Wilson[7,8], Yuk-Lam Ho ®[9] & Kelly Cho[9,10]

## Estonian Biobank Research Team

Kadri Arumäe[11], Uku Vainik[11,14,15], Andres Metspalu[14], Lili Milani[14], Tõnu Esko[14], Reedik Mägi[14], Mari Nelis[14] & Georgi Hudjashov[14]

