## [Peer Review File · Nature Communications]

Characterising the genetic architecture of changes in adiposity during adulthood using electronic health recordsREVIEWER COMMENTS

Reviewer #1 (Remarks to the Author):

Review report for “Characterizing the genetic architecture of changes in adiposity during adulthood using electronic health records”

Using UK biobank data, this paper addresses several key questions about the genetic architecture of adiposity changes during adulthood.

1. Quantified and replicated genetic contribution to the change of adiposity
2. Novel genetic findings with replication
3. Compelling sex-stratified analysis

It is a well-written and carefully designed study. I have several questions:

1. For both BMI and weight, in the first stage of LME modeling, do you need to also check the normality assumption? As LME assumptions are both error and random effects are normally distributed (equation 1).
2. LME adiposity changes analysis assumes a linear trend. The non-linear analysis learned several clusters that some of which clearly do not have a linear trend. This suggests model misspecification in the LME two-stage analysis. I am wondering how this impacts the analysis results, i.e., forcing a non-linear trend to have a linear trend, then using them for a GWAS analysis. Will this cause an inflated type I error, loss of power, or both?
3. Like in the clustering analysis, the uncertainty of slope estimation in LME cannot be taken into GWAS analysis. Also, Analysis cannot incorporate those who only have 1 measure, and cannot adjust for time-varying covariates. So I am curious whether a two-stage model for longitudinal data analysis in this GWAS setting when the sample size is large can overcome all the disadvantages in traditional longitudinal data analysis settings, i.e., SNP \times Time interaction analysis in LME
4. I wonder whether a polygenic risk score (PRS) of adiposity changes would make sense to be evaluated in addition to the current analysis.
5. All replication studies are within the UKB study – whether it is possible to seek an independent study to replicate these findings
6. Why adjust for 21 PC, and also use BOLT-LMM?
7. Figure 1A is dominated by published results – very hard to see novel/refined ones.

8. Biomarker phewas were conducted for rs429358, but why not phecode phewas ? whether this SNP is related to any cancer phenotypes which can influence BMI dramatically?

Reviewer #2 (Remarks to the Author):

Major comments

The article is very clearly written, with a logical presentation of the results. The topic is well-motivated and the results are robust. The methods are sound and solid, a great treatment of longitudinal data for GWAS. The follow-up analysis supporting the APOE association is convincing. Overall, I find it a great paper using the underexploited HER data for longitudinal analysis with very moderate genetic effects. Below, I list a couple of points, addressing them could improve the paper further.

The demonstrated gain in power using multiple measurements per individual is very elegant and a strong addition to the main message. It would be interesting to know how much variance per individual is observed in the BMI across time? How big is it relative to the BMI variance in the population? Does it correlate with the follow-up interval?

Comparison of the current findings and past results (<https://pubmed.ncbi.nlm.nih.gov/26426971>, <https://journals.plos.org/plosgenetics/article?id=10.1371/journal.pgen.1010303>) would be important.

The APOE (rs429358) variant strongly associates with lifespan/LDL/ALZ, which raises suspicion. I appreciate that the authors have included follow-up length to avoid survival bias. But this variant is correlated with age in the UKBB, due to survival bias. I'd suggest some extra analyses to ensure that the finding is not driven by survival bias: 1) Repeat the association for this SNP without correction for age+age². 2) Check other lifespan-associated SNPs (e.g. <https://pubmed.ncbi.nlm.nih.gov/29227965/>) for their association with the slope or the cluster memberships. If they do not show any signal, it is reassuring.

It seems to be a variant associated with changes in several traits and not always in the expected direction, based on its effect on BMI. What could be the biological explanation for this?

Given that 6 different (and close to independent) analysis were conducted [male/female x 3 memberships are all independent, on top of the slope analysis and the sex-combined], the multiple

testing should be corrected not only for the number of SNPs, but also the number of traits, thus a threshold of $1E-8$ would be fairer. This would remove 3 SNPs from Table 2... Related to this: why only the APOE variant is discussed and the other hits are quite neglected. All the follow-up tests presented in Fig 3 could have been tested for the 5 other SNPs of Table 2.

According to the UK Biobank approved project database there is no approved application with the number 10844 [<https://www.ukbiobank.ac.uk/enable-your-research/approved-research?query=10844#articles>]. Could you please double-check?

Minor comments

1. "Moreover, the genetic correlation between change in BMI and weight is nearly perfect" -> "Moreover, the genetic correlation between change in BMI and change in weight is nearly perfect"
2. In Eq (1) " $k = 1, 2$ " should be " $k = 0, 1$ ".
3. In Eq(2-3) $\epsilon_{0,i}$ should be $\epsilon_{i,0}$, same for $\epsilon_{1,i}$.
4. In Eq(2-3) hats are needed for the γ s.

Reviewer #3 (Remarks to the Author):

See attachment.

This is a nice methodologically sounded paper exploring the genetic architecture of changes in BMI vs BMI measured at 1 time point, which is what most of large GWASs have explored so far. I think the most important point of the paper is how difficult is to find genetic signals for longitudinal changes in BMI (and probably in many other traits). This is likely because the environmental influences play a larger role in this process. I feel this message can be highlighted more in the paper.

- Can you clarify which time period the EHR data are covering? I.e. what's the maximum and median follow-up time used when considering BMI measurements? I cannot easily find this info.
- With regard with follow-up time and the methods you used. How do you account for censoring? In other words, do you consider the beginning of follow-up as the first measurement and the end of follow-up the last measurement? Or do you consider the entire period covered by the EHR and account for censoring (e.g. death) ?
- How should one interpret the intercept from the LME? Would you get similar GWAS results if simply averaging all the BMI measurements and running a GWAS on that?
- The article would benefit from being clearer regarding the use of linear mixed models vs B-splines. Can the author provide a graphical illustration of which methods were used for which analysis? LME intercept was used for the BMI GWAS, while LME and B-spline + clustering were both used for the BMI-change GWAS. Correct? The LME was not use for clustering at all?
- Line 106 → “All analyses were adjusted for baseline obesity trait and confounders, including length of follow-up and number of follow-up measures, to mitigate survivor bias”. What are the “baseline obesity trait” ? Can you be more explicit about the rationale for adjusting for length of follow-up and number of follow-up measures? The two are also highly correlated and number of follow-ups correlates with BMI. What is the difference if you run a GWAS without using these covariates?
- APOE C allele from rs429358 increase mortality risk I would imagine, so these people should have a shorter follow-up and have less measurements, which might explain the BMI reduction over time. Can you check the association between rs429358 and the number of measurements? I understand you adjust for these effects in the model by controlling for length of follow-up and number of follow-up measures, but would be good to try to chase what exactly is going on with this APOE allele and how much this is explained by survival bias.
- Maybe you have done this analysis, but I missed it. But if you check all the SNPs that are GW-significant for BMI, is there an enrichment of signals for BMI-change? Or the two are completely independent?

- Please specify the genetic correlation between the LME slope approach and the clustering approach. In other words, how similar are the results between these two approaches ? (I guess $u_1 \text{ adj } u_0$ vs $\text{prob}(k_1) \text{ adj } u_0$)

- The genetic correlations between the BMI-change and the other measures are interesting, but can be better explained. Panel B of figure 4 is confusing. What are the correlation you are testing? Why the upper label says “genetic correlation with u_0 ” ? Then each panel seems to have another label. If I correctly understand the BMI-change phenotypes ($u_1 \text{ adj } u_0$ and $\text{prob}(k_1) \text{ adj } u_0$) have still a very high correlation with BMI at baseline, despite you having adjusted for that. Isn't then surprising that you don't see any of the GW-significant results for BMI from the largest GWAS of BMI popping up?

- I'm a little skeptical about the claim that heritability of weight change is higher in women than men. I can imagine many biases and the values are quite low anyway. Moreover, the distribution of the underlying phenotype is probably different so this might be difficult to make heritability estimates directly comparable. I would tone this down or seek for replication.

- Line 327, formula. Why did you use a log2 transformation?

- Formula (9) line 417, there should be and i' somewhere, right?

Reviewer 1

Using UK biobank data, this paper addresses several key questions about the genetic architecture of adiposity changes during adulthood.

1. Quantified and replicated genetic contribution to the change of adiposity
2. Novel genetic findings with replication
3. Compelling sex-stratified analysis

Authors' reply: Thank you for your comments – we very much appreciate you spending your valuable time assessing our research.

Comment 1.1 — For both BMI and weight, in the first stage of LME modeling, do you need to also check the normality assumption? As LME assumptions are both error and random effects are normally distributed (equation 1).

Authors' reply: Many thanks for this comment. We have now included Gaussian quantile-quantile plots of residuals and BLUPs in Supplementary Figures 10-12. The distributions of residuals and BLUPs can be seen to be somewhat heavy-tailed and/or skewed relative to a Gaussian. Such model misspecification could potentially lead to miscalibration of confidence intervals and hypothesis tests based on the standard linear mixed model (though this miscalibration would be mitigated by the large sample size, due to the central limit theorem). We therefore do not use or inspect any confidence intervals or hypothesis tests on the basis of the LME modelling itself. Rather, we take forward covariate-adjusted, inverse-normal transformed summary statistics to the stage-two GWAS software (BOLT-LMM), which then outputs well calibrated P-values, assuming any relevant confounding factors (e.g., population stratification) are adjusted for.

This discussion has been added to the manuscript text as below:

The distribution of residuals and BLUPs from the LME models are heavy-tailed relative to a Gaussian (Supp. Figs. 10-12). Such model misspecification could potentially lead to miscalibration of CIs and hypothesis tests based on the standard linear mixed model, although this is likely to be mitigated by the large sample size owing to the central limit theorem. We therefore take forward covariate-adjusted and inverse-normal transformed BLUPs, as described in (4), for genome-wide association testing. (page 15, lines 452-456)

Comment 1.2 — LME adiposity changes analysis assumes a linear trend. The non-linear analysis learned several clusters that some of which clearly do not have a linear trend. This suggests model misspecification in the LME two-stage analysis. I am wondering how this impacts the analysis results, i.e., forcing a non-linear trend to have a linear trend, then using them for a GWAS analysis. Will this cause an inflated type I error, loss of power, or both?

Authors' reply: Thanks for this perceptive comment. In the case of model misspecification where the linear trend model does not capture the true non-linear longitudinal behaviour, this can result in a loss of power relative to a model that captures the non-linear behaviour. This is because different clusters of non-linear shapes, should they exist, will be forced ineffectively into a one-dimensional representation (i.e. of slope) leading to lower signal-to-noise ratio (i.e. decreased information to resolve clusters as defined by non-linear shapes) and reduced power in that dimension-reduced space. However, such model misspecification will not lead to an inflated type I error. This is because the stage one-outputted summary statistics are covariate-adjusted and inverse-normal transformed prior to analysis in the stage-two BOLT-LMM software, meaning that type I error should be effectively controlled in stage two provided confounders are adjusted for. In summary, the effect you mention could result in loss of power, but should not result in inflated type I error.

Comment 1.3 — Like in the clustering analysis, the uncertainty of slope estimation in LME cannot be taken into GWAS analysis. Also, Analysis cannot incorporate those who only have 1 measure, and cannot adjust for time-varying covariates. So I am curious whether a two-stage model for longitudinal data analysis in this GWAS setting when the sample size is large can overcome all the disadvantages in traditional longitudinal data analysis settings, i.e., SNP×Time interaction analysis in LME

Authors' reply: Thank you for this interesting and thought-provoking comment. The desirable methodological properties that you mention are:

- (A) Admits propagation of **phenotypic uncertainty** (e.g. uncertainty of slope) to GWAS
- (B) Can incorporate information from individuals with only **one phenotypic measurement**
- (C) Accommodates adjustment for **time-varying covariates**
- (D) Handles a **large sample size** effectively
- (E) Effective in **GWAS setting** – large number of SNPs to test, population stratification, cryptic relatedness

The methods we wish to relate according to these desirable aspects are:

- (i) **One-stage traditional longitudinal LME**, i.e., SNP×Time interaction analysis in LME, fitted one SNP at a time, population stratification adjustment using principal component covariate adjustment
- (ii) **Two-stage linear-trajectory LME model**, taking LME intercept/slope forward to GWAS in BOLT-LMM
- (iii) **Two-stage nonlinear-trajectory clustering model**, clustering non-linear trajectories and taking forward soft (probabilistic) cluster assignments to GWAS in BOLT-LMM

We summarise the relationship between methods and properties in Table 3. Our overall assessment is that the major improvements offered by the two-stage methods over the traditional longitudinal analysis are: greater computational efficiency in dealing with large sample sizes (D); and the ability to flexibly incorporate the most cutting-edge approaches (e.g. BOLT-LMM) for efficiently analysing genome-wide SNP data and delivering high statistical power while adjusting for population stratification and cryptic relatedness (E). In terms of uncertainty propagation property (A), the non-linear clustering method is able to propagate uncertainty via the soft clustering assignment, while the two-stage linear trajectory approach is currently unable to propagate uncertainty forwards to GWAS (but it would be a beneficial future research direction to develop GWAS software that admits uncertainty propagation).

	(A)	(B)	(C)	(D)	(E)
(i)	Yes	No	Yes, in simple settings*	Slow	Slow
(ii)	Future research*	No	Yes, in simple settings	Yes	Yes
(iii)	Yes	No	Yes, in simple settings	Yes	Yes

Table 3: Assessment of one- and two-stage methods (i)-(iii) according to whether they have certain desirable properties (A)-(E), defined in the text of our response to Comment 1.3.

Comment 1.4 — I wonder whether a polygenic risk score (PRS) of adiposity changes would make sense to be evaluated in addition to the current analysis.

Authors' reply: Thank you for the suggestion to construct polygenic risk scores (PRSs) for adiposity-change, which may help predict an individual's genetic propensity to gain or lose weight. However, the predictive power of a PRS has a theoretical upper bound of the SNP-based heritability of the trait; in practice, uncertainty in GWAS effect-size estimates and differences between the base and target populations means that the predictive power of PRSs are typically much lower than heritability. Given that all of the adiposity-change traits we study

* In simple settings such as for exogenous time-varying covariates.

* It as an interesting avenue for future research to extend existing GWAS software to incorporate phenotypic inputs with accompanying measures of uncertainty.

here have very low SNP-based heritability ($< 3\%$), we believe that a PRS would be of limited additional value.

Comment 1.5 — All replication studies are within the UKB study – whether it is possible to seek an independent study to replicate these findings

Authors' reply: Thank you for this valuable suggestion that has greatly strengthened our results. We sought replication in two external cohorts: the Million Veterans Program based in the United States, and the Estonian Biobank. We were successful in replicating 9 of the 14 novel variants for baseline adiposity and 1 variant, rs429358, for adiposity change. We note that we were only sufficiently powered to seek replication for 4 of the 6 variants for adiposity change in the MVP cohort, and 1 variant in EstBB (see Supp. Table 25). That the remaining variants do not replicate may be because they are false positives, but we would also like to note the heterogeneity in population characteristics among these cohorts: participants in the MVP are on average 3.5 units of BMI heavier than those in the UKBB, predominantly male, and over-represent patients with cardiovascular disease and obesity⁴⁴; and participants in EstBB are on average 6 to 8 years younger than those in UKBB and are followed up for a mean of 4.7 years (as compared to 10-12 years in the UKBB). So while it is remarkable that the rs429358 effect reliably replicates across these cohorts, we would caution against the interpretation of null results for the other variants as false positives.

These results have been incorporated throughout the manuscript, but please particularly note the following sections of text in the Results:

Nine of the 14 novel SNPs replicate at $P < 3.6 \times 10^{-3}$ (FWER controlled at 5% across 14 tests using the Bonferroni method) in at least one of: (1) baseline obesity estimated with LME model intercepts in up to 437,703 individuals the MVP cohort, (2) baseline obesity estimated with LME model intercepts in up to 125,209 individuals the EstBB cohort, or (3) UKBB assessment centre measurements of cross-sectional obesity in up to 230,861 individuals not included in the discovery GWAS (Supp. Table 3) (page 4, lines 82-86)

and

The association of rs429358 with adiposity-change phenotypes was replicated at $P < 1.39 \times 10^{-3}$ (FWER controlled at 5% across six variants and six traits tested) in: (1) up to 437,703 individuals in the MVP cohort, and (2) up to 125,209 individuals in the Estonian Biobank, and (3) up to 17,035 individuals in UKBB with multiple measurements of weight and BMI at repeat assessment centre visits who were excluded from the discovery analyses (Figure 4 and Supp. Table 5). (page 6, lines 132-136)

We have also added relevant sections to the Methods and Discussion.

Comment 1.6 — Why adjust for 21 PC, and also use BOLT-LMM?

Authors' reply: As the reviewer indicates, BOLT-LMM runs a mixed model that accounts for population structure in the model, whereas standard linear regression requires adjustment for principal components (PCs) to account for population stratification. However, when heritability estimate approaches zero, linear mixed models degenerate to simple linear regression and hence BOLT-LMM requires the additional adjustment for PCs⁸⁷.

Comment 1.7 — Figure 1A is dominated by published results – very hard to see novel/refined ones.

Authors' reply: Thank you for this comment; we assume you are referring to Figure 2A, which is the Manhattan plot for baseline adiposity associations that shows reported, refined, and novel SNPs, as Figure 1A displays model fits for a selection of individuals with adiposity data and has no published/novel/refined classification. We have adjusted the point-size of the novel/refined variants in the Manhattan plot to make

them more visible, which we hope addresses the issue. We have also added Supp. Fig. 13, which only shows the novel/refined variants for further clarity.

Comment 1.8 — Biomarker phewas were conducted for rs429358, but why not phecode phewas ? whether this SNP is related to any cancer phenotypes which can influence BMI dramatically?

Authors' reply: Thank you for this suggestion. As the association of rs429358 with various phenotypes has been widely studied, including phenome-wide association analyses at sample sizes larger than those available in UK Biobank⁶⁴, we did not attempt to duplicate these efforts and instead focussed on the longitudinal biomarker associations (i.e. association of rs429358 with change in biomarker levels over time), which has never been reported previously. However, at the reviewer's request, we have now included a supplementary figure with the PheWAS conducted in UKBB and added descriptive text (Supp. Fig. 14). This SNP is not associated with any of the 38 cancer phenotypes at phenome-wide significance ($P < 0.05/290$ phenotypes tested), so we did not perform any additional sensitivity analyses that exclude individuals with cancer diagnoses from our results. The following text has been updated in the manuscript:

*The APOE locus is a highly pleiotropic region that is associated with lipid levels^{58,59}, Alzheimer's disease^{60,61}, and lifespan^{62,63}, among other traits⁶⁴, both in the UKBB (**Supp. Fig. 14**) and elsewhere. (page 6, lines 161-162)*

Reviewer 2

Reviewer's major comments

Comment 2.1 — The article is very clearly written, with a logical presentation of the results. The topic is well-motivated and the results are robust. The methods are sound and solid, a great treatment of longitudinal data for GWAS. The follow-up analysis supporting the APOE association is convincing. Overall, I find it a great paper using the underexploited HER data for longitudinal analysis with very moderate genetic effects. Below, I list a couple of points, addressing them could improve the paper further.

Authors' reply: Thank you for your kind words and positive assessment our work – we very much appreciate you dedicating your valuable time to review our research.

Comment 2.2 — The demonstrated gain in power using multiple measurements per individual is very elegant and a strong addition to the main message. It would be interesting to know how much variance per individual is observed in the BMI across time? How big is it relative to the BMI variance in the population? Does it correlate with the follow-up interval?

Authors' reply: Thanks for this interesting comment! The intra-individual variance in BMI (mean = 3.02) is about ten times lower than the inter-individual variance (30.3), a difference that is consistent across men and women (see tables/RR-2-2.xlsx). The same holds for intra-individual vs population variance in weight, with the former being about 10x lower. There is a large spread in the values of intra-individual variance (figs/RR-2-2-distributions.png), and the value of this variance is mildly positively correlated with follow-up metrics (R2 between intra-individual variance in BMI and length of follow-up = 0.111, R2 with number of follow-up measures = 0.0581) (tables/RR-2-2.xlsx). However, the considerable heteroskedasticity prevents inference on the exact nature of these relationships (figs/RR-2-2-correlations.png). We also performed joint modelling using the TrajGWAS software to test for genetic variants associated with the longitudinal mean and within-individual variance in weight, but confirm previous findings of no genetic associations with intra-individual variance in weight (Supp. Fig. 15). Results for these analyses have been added as follows:

Intra-individual variance is another longitudinal metric of interest, however we (Supp. Fig. 15) and others⁵³ find no genetic variants associated with intra-individual variance in weight over time. (page 4, lines 93-94)

and methods:

Analyses were performed using the TrajGWAS package⁵³ in Julia¹³⁶, for 177,472 unrelated individuals of white British ancestry with multiple measurements of weight included in the discovery analyses. Briefly, TrajGWAS analysis is conducted in two stages to test for genetic effects on longitudinal trajectory mean, intra-individual variance, and a joint effect on either mean or variance in a LME model framework⁵³. In the first stage, we fit a null model for weight with fixed effects for the intercept, age, age², sex, and 21 genetic PCs; we included random effects for the intercept and linear slope of age. In the second stage, we performed score testing with the saddle-point approximation under the full model, i.e. including genome-wide effects for all variants with MAF > 1% in the genotyped and imputed UKBB data that passed QC. (page 25, lines 713-720)

Comment 2.3 — Comparison of the current findings and past results (<https://pubmed.ncbi.nlm.nih.gov/26426971>, <https://journals.plos.org/plosgenetics/article?id=10.1371/journal.pgen.1010303>) would be important.

Authors' reply:

Thank you for the suggestion. We now include the following text in our Discussion:

Previous studies have estimated continuity in the genetic correlation of BMI measured at different ages⁸¹, which is theorised to emerge by two possible mechanisms⁸²: (1) common genetic (or environmental) factors are associated with the rates of change in BMI over time, which we test in this study, and (2) that these correlations are induced by time-specific genetic (or environmental) factors in an autoregressive manner, i.e. BMI genetics at time-point $t - 1$ causally affect BMI at time t . Studies testing the latter hypothesis have arrived at opposing conclusions: Gillespie et al. (2022)⁸³ find that on a genome-wide scale, age-specific genetic effects in an autoregressive framework do not explain differences in BMI heritability across ages 40-73 years, while Winkler et al. (2015) did identify 15 genetic loci with differential effects on BMI in younger adults (age < 50 years) and older adults (age > 50 years). Both studies were "pseudo-longitudinal", i.e. the same individuals were not monitored over a period of time, but rather cross-sectional individual data was grouped into age-bins. Our work tests a distinct hypothesis and is also, to our knowledge, the first to perform a truly longitudinal genetic study with repeated measures in this age group. (page 9, lines 245-256)

Comment 2.4 — The APOE (rs429358) variant strongly associates with lifespan/LDL/ALZ, which raises suspicion. I appreciate that the authors have included follow-up length to avoid survival bias. But this variant is correlated with age in the UKBB, due to survival bias. I'd suggest some extra analyses to ensure that the finding is not driven by survival bias: 1) Repeat the association for this SNP without correction for age+age². 2) Check other lifespan-associated SNPs (e.g. <https://pubmed.ncbi.nlm.nih.gov/29227965/>) for their association with the slope or the cluster memberships. If they do not show any signal, it is reassuring.

Authors' reply:

Thank you for this insightful comment. We have strengthened the analyses against survival bias even further by following your suggestions. The following text has been added to the Results:

Despite the association of rs429358 with lifespan, we found no association between this variant and follow-up metrics in our study (Supp. Table 22); we also found no significant difference in the effect of this variant on adiposity change from two sets of models: (1) without including age and related covariates, i.e. follow-up metrics and year of birth, and (2) with these covariates (heterogeneity $P_{het} < 0.05$) (Supp. Fig. 16). Finally, we observe no associations between 135 of 138 published lifespan-associated genetic variants and our adiposity-change phenotypes at $P < 3.6 \times 10^{-4}$ (FWER controlled at 5% across 138 tests via the Bonferroni method). Of the three SNPs associated with both weight change and lifespan, two (rs429358 and rs7412) are variants in the APOE gene, and rs1085251 is a known obesity association in the FTO locus (Supp. Table 16). (page 7, lines 165-173)

and the accompanying sections in the Methods:

We curated a list of 138 independent variants associated with longevity in the GWAS Catalog⁵⁵, accessed on 27 March 2023 (Supp. Table 16). We identified independent SNPs that passed genotyping and imputation QC filters in UKBB by pairwise pruning variants in LD ($r^2 > 0.1$) within a 1 Mb window. One of the lead variants identified in this study, i.e. rs429358 in the APOE locus, was pruned out in favour of rs4420638, which is 11 kb away from the lead variant and in LD with rs429358 with $r^2 = 0.69$. We looked up the effects of these variants in the various adiposity-change GWAS summary statistics and established significance at $P = 3.60 \times 10^{-4}$ (Bonferroni-corrected at 5% across 138 tests). (page 25, lines 699-705)

and Discussion:

As longevity may confound the APOE-weight loss association^{62, 63}, we adjusted analyses for the length of follow-up in EHR to mitigate against survivor bias; however, we also present age-unadjusted analyses and demonstrate that other lifespan-associated variants are not associated with adiposity change in our GWASs. (page 11, lines 322-325)

have also been updated.

Comment 2.5 — It seems to be a variant associated with changes in several traits and not always in the expected direction, based on its effect on BMI. What could be the biological explanation for this?

Authors' reply: Thank you for this comment, as you say, the *APOE* rs429358 variant is highly pleiotropic⁶⁴. While some of the longitudinal effects we report here, particularly the lipid markers, are likely to be tightly linked to weight-change, others may be independent of this association. For example, in our longitudinal analysis, the BMI-lowering 'C' allele of rs429358 is associated with a reduction in triglyceride levels and increase in HDL-cholesterol levels over time. These are expected directions of effect given the literature on changes in lipid levels with weight change⁹⁵. On the other hand, the 'C' allele of rs429358 is associated with higher cross-sectional triglyceride levels⁹⁶ and lower HDL-cholesterol levels⁹⁵, which are the opposite of what one would expect given its BMI association. We have summarised the longitudinal and cross-sectional associations of rs429358, along with the observed phenotypic effects of BMI on each trait, in **Table RR-2-5.xlsx**. In all, we believe it is not straightforward to hypothesise the effect of a single (pleiotropic) variant on polygenic traits, such as BMI and cholesterol, based on the correlations between phenotypes alone. Teasing apart the precise causal pathways (or lack of causal pathways) between these various effects is beyond the scope of this manuscript. We have added a sentence to the Discussion to reflect these unexpected associations:

Some of the effects of rs429358 are discordant with previously reported phenotypic correlations between obesity and these biomarkers, however, the causal longitudinal and pleiotropic nature of these associations remain to be established. (page 11, lines 318-320)

Comment 2.6 — Given that 6 different (and close to independent) analysis were conducted [male/female × 3 memberships are all independent, on top of the slope analysis and the sex-combined], the multiple testing should be corrected not only for the number of SNPs, but also the number of traits, thus a threshold of 1E-8 would be fairer. This would remove 3 SNPs from Table 2... Related to this: why only the *APOE* variant is discussed and the other hits are quite neglected. All the follow-up tests presented in Fig 3 could have been tested for the 5 other SNPs of Table 2.

Authors' reply: Thank you for this suggestion. As the adiposity-change metrics we study are all highly correlated, the tests we perform are not fully independent of each other; however, we acknowledge this multiple testing burden and have marked the three lead SNPs in Table 2 that would not pass a Bonferroni-adjusted threshold of $P < 1E-8$ as suggested. We also conducted the follow-up analyses presented for rs429358 in the *APOE* locus, i.e. association with abdominal adiposity change, and association with change in 45 other biomarkers over time, for all 6 lead SNPs reported in Table 2.

Brief results:

1. Other than rs429358, no other lead SNP is associated with self-reported weight change in the past year (all $P > 0.01$).
2. At $P < 0.002$ (accounting for 4 adiposity-change metrics and 6 lead SNPs), the reported rs429358 associations with WC-loss and WHR-loss remain significant; in addition, we find that the weight-increasing allele of rs9467663 is also associated with gain in WCadjBMI over time ($\beta = 0.0214$).
3. At $P < 2 \times 10^{-4}$ (accounting for 45 biomarkers and 6 lead SNPs), 6/7 of the reported rs429358 associations (with change over time in cholesterol, CRP, HDL cholesterol, lymphocytes, potassium, and triglyceride) remain significant, with the exception of haemoglobin concentration, which we have now removed from the main figure. In addition, we find that the weight-increasing allele of rs9467663 is associated with gain over time in iron ($\beta = 0.176$), haematocrit packed cell volume % ($\beta = 0.00652$), and mean corpuscular volume ($\beta = 0.0145$), and loss over time in haemoglobin concentration ($\beta = -0.0233$). Similarly, the BMI-increasing allele of chr6:26076446 is associated with gain over time in iron ($\beta = 0.133$) and loss over time in haemoglobin concentration ($\beta = 0.0144$). Both of these SNPs have been previously reported to associate with cross-sectional values of haematological traits.

A brief discussion of these results has been added to the main text and presented in supplementary tables (**Supp. Tables 17-19**):

Other than rs429358, none of the lead variants for adiposity change replicated in either MVP or EstBB at $P > 1.39 \times 10^{-3}$ (FWER controlled at 5% across 6 variants via the Bonferroni method) (Supp. Table 5). However, we were only sufficiently powered to replicate the effects of

three of these in MVP (*rs9467663*, *chr6:26076446*, and the male-specific variant *rs12953815*), and none in *EstBB*, as replication at 80% power required sample sizes of between 116,000 to 234,000 individuals with repeat measurements of BMI (**Supp. Table 25**). (page 7, lines 187-192)

Comment 2.7 — According to the UK Biobank approved project database there is no approved application with the number 10844 <https://www.ukbiobank.ac.uk/enable-your-research/approved-research?query=10844#articles>]. Could you please double-check?

Authors' reply: Thanks for spotting this, it appears to be an unintentional mistake on our part. This should read application number 11867.

Reviewer's minor comments

Comment 2.8 — “Moreover, the genetic correlation between change in BMI and weight is nearly perfect”
-> “Moreover, the genetic correlation between change in BMI and change in weight is nearly perfect”

Authors' reply: Thank you for the clarification, we have edited this now:

Moreover, the genetic correlation between change in BMI and weight is nearly perfect (page 8, lines 219-220)

Comment 2.9 — In Eq (1) “ $k = 1, 2$ ” should be “ $k = 0, 1$ ”.

Authors' reply: Many thanks for spotting this, now corrected at (1).

Comment 2.10 — In Eq(2-3) $\epsilon_{0,i}$ should be $\epsilon_{i,0}$, same for $\epsilon_{1,i}$.

Authors' reply: Thank you, now corrected in (2).

Comment 2.11 — In Eq(2-3) hats are needed for the γ s.

Authors' reply: Thanks for pointing this out, we have now improved the notation in that section and clarified that the γ_0 and γ_1 in (2) and (3) are distinct from the previously defined γ (and hence do not require hats in (2) and (3)):

The coefficient vectors γ_0 and γ_1 in (2) and (3) are estimated by least squares and are distinct from the previously estimated γ in (1). (page 15, lines 446-447)

Reviewer 3

Reviewer's comments

Comment 3.1 — This is a nice methodologically sounded paper exploring the genetic architecture of changes in BMI vs BMI measured at 1 time point, which is what most of large GWASs have explored so far. I think the most important point of the paper is how difficult is to find genetic signals for longitudinal changes in BMI (and probably in many other traits). This is likely because the environmental influences play a larger role in this process. I feel this message can be highlighted more in the paper.

Authors' reply: Thank you for your positive assessment of our work, and for dedicating your valuable time to reviewing our paper. Our results suggest that it is relatively difficult to identify genetic associations with longitudinal changes in obesity traits, compared with identifying loci associated with cross-sectional BMI (e.g., a single or averaged measurement). Intuitively, SNPs exhibiting association with cross-sectional BMI must have had a causal impact on expected longitudinal BMI at some periods in individuals' lifespans; for example, the typical BMI SNP risk allele might have a life-long, constant, and weak effect on BMI, whereas some SNPs might act strongly, but only in early life.

In performing genetic association with cross-sectional BMI across a population, the phenotype captures the cumulative longitudinal effects of each particular BMI SNP genotype up to the age at which each individual is measured. In contrast, our derived measures of longitudinal change essentially estimate the rate of change of BMI over a *relatively short* time period after the date of first measurement of an individual. The magnitude of genetic signal thus tends to be smaller in the longitudinal analysis compared to the cross-sectional one. In contrast, some of the environmental processes are of the same magnitude in both cases, particularly short-term high-frequency environmental variation occurring over the space of a few years. This can lead to smaller signal-to-noise ratio, and therefore relatively low power to detect genetic effects in longitudinal analyses.

More broadly, there are several factors that might affect the relative power to detect longitudinal effects such as: sample size, typically being smaller in longitudinal studies; the longer and more frequent the typical follow-up is in a longitudinal study, the greater the power; and the particular statistical methods used to estimate cross-sectional or longitudinal traits can affect the accuracy and precision of estimates, and hence the strength of genetic signal detected.

We have added the following text to the Discussion:

However, our results indicate the relative difficulty of identifying genetic associations with longitudinal changes in obesity traits, compared with identifying loci associated with cross-sectional BMI. Variants associated with cross-sectional BMI must have had a causal impact on expected longitudinal BMI at some periods in individuals' lifespans; i.e. a cross-sectional BMI phenotype captures the cumulative longitudinal effects of each BMI-associated genotype up to the age at which the individual is measured. In contrast, our derived measures of longitudinal change target the rate of change of BMI over a shorter average time period, and the magnitude of genetic signal thus tends to be smaller in the longitudinal analysis compared to the cross-sectional one. This means that the weaker longitudinal genetic signal can be obscured by the non-genetic contribution from individuals' short-and long-term environment, whilst the stronger cross-sectional genetic signal may be detected with higher power as the signal-to-noise ratio is larger. More broadly, there are several factors that might affect the relative power to detect longitudinal effects such as: sample size, typically being smaller in longitudinal studies; the longer and more frequent the typical follow-up is in a longitudinal study, the greater the power; and the particular statistical methods used to estimate cross-sectional versus longitudinal traits can affect the accuracy and precision of estimates, and hence the strength of genetic signal detected. (page 10, lines 295-309)

Comment 3.2 — Can you clarify which time period the EHR data are covering? I.e. what's the maximum and median follow-up time used when considering BMI measurements? I cannot easily find this info.

Authors' reply: Thank you for the comment - the median follow-up time is presented in Table 1; we have included minimum and maximum values for your reference in tables/RR-3-2.xlsx.

Comment 3.3 — With regard with follow-up time and the methods you used. How do you account for censoring? In other words, do you consider the beginning of follow-up as the first measurement and the end of follow-up the last measurement? Or do you consider the entire period covered by the EHR and account for censoring (e.g. death)?

Authors' reply: Thanks for your perceptive comment. In short, we do consider the beginning of follow-up to be the first measurement and the end of follow-up to be the last measurement, and we linearly adjust for some observed covariates that may affect when, and how often, an individual get measured (referred to as their *measurement process* below) and/or whether an individual is included in the study (referred to as their *study selection process* below).

To address your comment in more detail, in this dataset there are many factors that affect the timing and availability of observations for any particular individual, including your example of censoring by death; an additional example of censoring would be the process by which an individual is cured of a disease, leading to discontinuation of measurement in primary care. There are several additional factors that affect measurement, for example: (self)-selection to participate in UKBB may be causally driven by an individual's characteristics; similarly, the presence of BMI measurements in primary care records may be causally driven by an individual's medical conditions.

For our genetic association studies, we can think of the treatment (SNP genotype) as having been randomly allocated to individuals. Further, any particular common SNP will have negligible causal effect on any complex trait, in particular on any traits linked to the study selection/measurement/censoring processes.* Therefore, since the study selection/measurement/censoring processes are independent of the treatment (SNP genotype), they do not act as confounders, and we should obtain unbiased estimates of average treatment (genetic) effects (intuitively, the study selection/measurement/censoring processes are the same in all genotypic classes).

While these genetic effects are estimated in the measured subpopulation (the UKBB), they should apply to the wider UK population, unless there is a gene-environment interaction. In the case of gene-environment interaction, if the phenotypic "environment" is causally linked to censoring/study inclusion/measurement processes, then the genetic effects estimated (under a model without an explicit interaction term) may differ from those of the wider UK population; e.g., if a SNP acts more strongly in 45-65 year-olds than in other age groups, then our model would have an upwardly biased estimate of the magnitude of genetic effect compared to the general UK population (as UKBB is enriched with 45-65 year-olds relative to the general population).

Comment 3.4 — How should one interpret the intercept from the LME? Would you get similar GWAS results if simply averaging all the BMI measurements and running a GWAS on that?

Authors' reply: Thank you for this insightful comment. The intercept for an individual should be interpreted as a (scaled) estimate of that individual's obesity traits at the age of first measurement in our UKBB and primary care combined dataset. The scaling is performed across the entire subpopulation represented in dataset via a rank-based inverse normal transformation, so it is useful to consider it approximately as representing SDs from the subpopulation mean.

As expected, average BMI and weight are highly correlated with the intercepts from the LME ($R^2 = 0.95$), see tables/RR-3-4.xlsx. As suggested, we ran a GWAS on the rank-based inverse normal transformed average adiposity trait (BMI or weight) within each sex strata (female-specific, male-specific, or sex-combined), with the same adjustment for covariates as in the LMM intercept GWAS. We find that between 80-90% of associations discovered in the average-trait GWAS are also genome-wide significant (GWS, $P < 5E-08$) in the LMM intercept GWAS (tables/RR-3-4.xlsx), and the effect sizes of GWS SNPs are practically identical in both studies ($R^2 = 0.998$) (see figs/RR-3-4.png). It is worth noting that the LMM intercept GWAS appears better powered to identify associations, despite identical sample sizes, as we discover up to 1.2x more GWS SNPs associated with LMM intercept than with average trait in the same strata (tables/RR-3-4.xlsx).

We have added a brief comment on these to the Results:

*There is potentially the confounding effect of population stratification, with spatial location as a confounding variable, but we deal with this within the methodological framework of BOLT-LMM.

While the within-individual mean and baseline trait modelled from LME are phenotypically ($R^2 > 0.95$) and genetically highly correlated ($R^2 > 0.99$) (Supp. Fig. 17), the LME intercept appears better powered for genetic association testing than the average trait, as we discover up to 1.2x more GWS variants associated with the former (Supp. Table 20). (page 5, lines 94-98)

Comment 3.5 — The article would benefit from being clearer regarding the use of linear mixed models vs B-splines. Can the author provide a graphical illustration of which methods were used for which analysis? LME intercept was used for the BMI GWAS, while LME and B-spline + clustering were both used for the BMI-change GWAS. Correct? The LME was not use for clustering at all?

Authors' reply: Thank you for this suggestion. Your interpretation is correct and we have now included in the Supplementary Information a workflow for carrying forward model estimates to GWAS (Supp. Fig. 18).

Comment 3.6 — Line 106: "All analyses were adjusted for baseline obesity trait and confounders, including length of follow-up and number of follow-up measures, to mitigate survivor bias". What are the "baseline obesity trait"? Can you be more explicit about the rationale for adjusting for length of follow-up and number of follow-up measures? The two are also highly correlated and number of follow-ups correlates with BMI. What is the difference if you run a GWAS without using these covariates?

Authors' reply: Thanks for the clarification; the "baseline obesity trait" refers to the first measure of the trait (BMI or weight) for each individual in our data. As you say, this is indeed modestly positively correlated with follow-up metrics (length and number), which we note in lines:

On average, women with ten or more weight measurements are 8.3 kg (3.7 units of BMI) heavier than their counterparts with 1-3 measurements; for men, this is an 8.2 kg (3.1 units of BMI) difference. (page 5, lines 100-102)

Length and number of follow-up metrics may also indicate survivor bias, although the direction of association is not straightforward - unhealthier individuals visit the GP more, and are weighed more frequently; however, healthier individuals who survive longer may also have longer follow-up in our records. Moreover, the length and number of follow-ups also affects the uncertainty of estimates generated by our models for adiposity-change. We attempted to imperfectly account for these effects via the adjustment for length and number of follow-ups.

To see how the adjustment influences genetic associations, we conducted GWAS for linear slope change in adiposity without adjusting for length and number of follow-up metrics as suggested. rs429358 still emerges as the only lead SNP across all strata, and no additional loci reach genome-wide significance in the GWAS unadjusted for length of follow-up and number of follow-up measures. As seen in figs/RR-3-6.png, there is also no difference in effect size of rs429358 on slope-change in adiposity between the follow-up-adjusted and unadjusted studies.

Further, we compared the GWASs for probabilities of belonging to various adiposity change clusters adjusted for follow-up metrics vs unadjusted for these metrics (tables/RR-3-6.xlsx). All lead SNPs discovered in the original GWASs (adjusted for follow-up metrics) were near GWS ($P < 5E-07$) in the unadjusted analyses. Similarly, any lead SNPs newly identified in the unadjusted GWASs also achieved $P < 5E-07$ in the original GWASs. The only exception to this are lead SNPs in the *FTO* locus, which are identified in the male-specific and sex-combined GWASs for probabilities of belonging to any of the top three adiposity clusters (high-gain, moderate-gain, and stable) only in the unadjusted analysis. *FTO* is among the strongest known associations with obesity¹³⁷, and is also among the strongest associations with baseline adiposity in our analyses, suggesting that this association is likely driven by baseline adiposity rather than adiposity-change. The adjustment for length of follow-up and number of follow-up metrics in the original analyses likely accounted for this effect due to correlation with baseline adiposity.

A brief report of these results has been added:

*While all lead variants in the discovery GWASs remain significant at $P < 5 \times 10^{-7}$ in GWASs that are not adjusted for follow-up metrics, we discover three variants in the *FTO* locus that are*

associated with BMI or weight gain only in analyses that are unadjusted for follow-up metrics (Supp. Table 21). These associations may reflect genetic contributions to baseline weight rather than weight-change, as FTO is among the strongest known loci for obesity and follow-up metrics are strongly positively correlated with baseline obesity (Supp. Table 4). (page 7, lines 192-197)

Comment 3.7 — APOE C allele from rs429358 increase mortality risk I would imagine, so these people should have a shorter follow-up and have less measurements, which might explain the BMI reduction over time. Can you check the association between rs429358 and the number of measurements? I understand you adjust for these effects in the model by controlling for length of follow-up and number of follow-up measures, but would be good to try to chase what exactly is going on with this APOE allele and how much this is explained by survival bias.

Authors' reply: Thank you for the insightful comment, as there could plausibly be an association between rs429358, which is associated with survival, and follow-up metrics in our data. We conducted a linear regression to test the effect of each additional copy of the minor 'C' allele of rs429358 on number of adiposity metrics and length of follow-up (in years) in our data (tables/RR-3-7.xlsx). At $P < 0.05$, there is no effect of rs429358 'C' allele dosage on the length of follow-up; however, each additional copy of this allele is associated with an average reduction of 0.108 (SE = 0.048, $P = 0.026$) measures of BMI in women, and 0.122 (SE = 0.052, $P = 0.019$) measures of weight in women. Although the direction of effect is consistent in men, there is no significant association between rs429358 allele dosage and number of follow-up metrics in men (all $P > 0.5$). It is therefore plausible, but unlikely, that survival bias substantially modifies the association between rs429358 and adiposity-change. Indeed, as we show in figs/RR-3-6.png, the associations between rs429358 and all adiposity-change metrics are virtually unchanged upon adjustment for follow-up metrics.

We summarise the results from this and other analyses to examine possible survival bias effects of rs429358 on adiposity-change:

Despite the association of rs429358 with lifespan, we found no association between this variant and follow-up metrics in our study (Supp. Table 22); we also found no significant difference in the effect of this variant on adiposity change from two sets of models: (1) without including age and related covariates, i.e. follow-up metrics and year of birth, and (2) with these covariates (heterogeneity P -value $P_{het} < 0.05$) (Supp. Fig. 16). Finally, we observe no associations between 135 of 138 published lifespan-associated genetic variants and our adiposity-change phenotypes at $P < 3.6 \times 10^{-4}$ (FWER controlled at 5% across 138 tests via the Bonferroni method). Of the three SNPs associated with both weight change and lifespan, two (rs429358 and rs7412) are variants in the APOE gene, and rs1085251 is a known obesity association in the FTO locus (Supp. Table 16). (page 7, lines 165-173)

Comment 3.8 — Maybe you have done this analysis, but I missed it. But if you check all the SNPs that are GW-significant for BMI, is there an enrichment of signals for BMI-change? Or the two are completely independent?

Authors' reply: Thank you for this comment. Variants that are associated with BMI do indeed have higher χ^2 statistics for association with BMI-change than would be expected by chance, which we observe in QQ-plots across all strata (Supp. Figure 19). As you suggest, this is because the traits are not independent. We have added the following text to the Results to reflect this:

As expected given their positive correlation, we observe inflation of the χ^2 statistics for adiposity-change slope associations amongst lead variants for baseline adiposity (Supp. Figure 19). (page 8, lines 210-211)

Comment 3.9 — Please specify the genetic correlation between the LME slope approach and the clustering approach. In other words, how similar are the results between these two approaches? (I guess $u1 \text{ adj } u0$ vs $\text{prob}(k1) \text{ adj } u0$)

Authors' reply: Thanks for the suggestion: as expected, and reassuringly, the genetic correlation between $u1$ -adj- $u0$ (linear slope from LME) and $p(k1)$ -adj- $u0$ (probability of belonging to the high-gain cluster from non-linear modelling approach) is nearly perfect in all strata (tables/RR-3-9.xlsx).

Comment 3.10 — The genetic correlations between the BMI-change and the other measures are interesting, but can be better explained. Panel B of figure 4 is confusing. What are the correlation you are testing? Why the upper label says "genetic correlation with $u0$ " ? Then each panel seems to have another label. If I correctly understand the BMI-change phenotypes ($u1$ adj $u0$ and $\text{prob}(k1)$ adj $u0$) have still a very high correlation with BMI at baseline, despite you having adjusted for that. Isn't then surprising that you don't see any of the GW-significant results for BMI from the largest GWAS of BMI popping up?

Authors' reply: Thank you for this note. In this figure (old: Figure 4, revised: Figure 5) we wanted to demonstrate that the baseline adiposity traits ($u0$) are correlated with the adiposity-change traits (linear slope: $u1$, and probability of belonging to the high-gain cluster ($p(k1)$). We have modified the figure labels and legend to clarify. You also correctly note that the baseline BMI and BMI-change traits are highly correlated, and as we now show in **Supp. Figure 19** (also in response to Comment 3.8), the BMI-change GWASs are enriched for baseline BMI variants. None of these attain genome-wide significance, but this may be a limitation of the sample size and it is likely that some BMI variants may be associated with BMI-change with effect sizes that were too small to detect in our study.

Comment 3.11 — I'm a little skeptical about the claim that heritability of weight change is higher in women than men. I can imagine many biases and the values are quite low anyway. Moreover, the distribution of the underlying phenotype is probably different so this might be difficult to make heritability estimates directly comparable. I would tone this down or seek for replication.

Authors' reply: Thank you very much for the comment. In the Estonian Biobank, where we sought replication, we estimate heritability of BMI-change in females to be 0.0215 (SE = 0.0056), and of males to be 0.018 (SE = 0.0098). These values are low, but they are similar to those we estimate in the UK Biobank discovery analyses (female $h^2 = 0.0289$ (SE = 0.0056), and male $h^2 = 0.0105$ (SE = 0.0059)). However, as you mention, it is difficult to directly compare these due to differences in the underlying phenotype, so we have clarified this in the text, removed the "sex-specific" label from the relevant Results section header, and removed the phrase in the Abstract referencing higher heritability of adiposity-change in women:

Furthermore, we observe that the heritability of BMI and weight trajectories are higher in women than in men (2.89% (0.56) vs 1.05% (0.59) for BMI slopes, $P_{sexhet} = 0.012$; and 3.42% (0.53) vs 1.69% (0.52) for weight slopes, $P_{sexhet} = 9.9 \times 10^{-3}$). Similarly, we estimate the heritability of BMI slopes in the Estonian Biobank to be higher in women (2.15% (0.56) in women vs 1.80% (0.98) in men); however, these values are low and must be interpreted with caution. (page 8, lines 202-207)

Comment 3.12 — Line 327, formula. Why did you use a log2 transformation?

Authors' reply: Thank you for the note. We used a log transformation to reduce skewness in the distribution of our defined jump $P_{i,j}$. We used base 2 in particular so that we could readily interpret changes of one unit on the log scale as a two-fold change, which was useful for our internal discussions around choice of our quality control jump threshold (i.e., where we used ± 3 population SD on centred $P_{i,j}$). The scale of $P_{i,j}$ is thus log2 of proportional change in BMI per unit time. The SD of the $P_{i,j}$ is 2.22. So, the interpretation of our upper 3SD threshold is as a multiplicative increase of $2^{3 \times 2.22} \approx 101$ from the geometric mean of the proportional change in BMI per unit time.

Comment 3.13 — Formula (9) line 417, there should be and i' somewhere, right?

Authors' reply: Thank you for spotting this typo, which we have now corrected.

References

- ¹ Bluher, M. Obesity: global epidemiology and pathogenesis. *Nat Rev Endocrinol* **15**, 288–298 (2019). URL <https://www.ncbi.nlm.nih.gov/pubmed/30814686>.
- ² Collaborators, G. B. D. O. *et al.* Health effects of overweight and obesity in 195 countries over 25 years. *N Engl J Med* **377**, 13–27 (2017). URL <https://www.ncbi.nlm.nih.gov/pubmed/28604169>.
- ³ Must, A. *et al.* The disease burden associated with overweight and obesity. *JAMA* **282**, 1523–9 (1999). URL <https://www.ncbi.nlm.nih.gov/pubmed/10546691>.
- ⁴ Loos, R. J. F. & Yeo, G. S. H. The genetics of obesity: from discovery to biology. *Nat Rev Genet* **23**, 120–133 (2022). URL <https://www.ncbi.nlm.nih.gov/pubmed/34556834>.
- ⁵ Maes, H. H., Neale, M. C. & Eaves, L. J. Genetic and environmental factors in relative body weight and human adiposity. *Behav Genet* **27**, 325–51 (1997). URL <https://www.ncbi.nlm.nih.gov/pubmed/9519560>.
- ⁶ Elks, C. E. *et al.* Variability in the heritability of body mass index: a systematic review and meta-regression. *Front Endocrinol (Lausanne)* **3**, 29 (2012). URL <https://www.ncbi.nlm.nih.gov/pubmed/22645519>.
- ⁷ Khera, A. V. *et al.* Polygenic prediction of weight and obesity trajectories from birth to adulthood. *Cell* **177**, 587–596 e9 (2019). URL <https://www.ncbi.nlm.nih.gov/pubmed/31002795>.
- ⁸ Hardy, R. *et al.* Life course variations in the associations between fto and mc4r gene variants and body size. *Hum Mol Genet* **19**, 545–52 (2010). URL <https://www.ncbi.nlm.nih.gov/pubmed/19880856>.
- ⁹ Silventoinen, K. *et al.* Changing genetic architecture of body mass index from infancy to early adulthood: an individual based pooled analysis of 25 twin cohorts. *Int J Obes (Lond)* **46**, 1901–1909 (2022). URL <https://www.ncbi.nlm.nih.gov/pubmed/35945263>.
- ¹⁰ Helgeland, O. *et al.* Characterization of the genetic architecture of infant and early childhood body mass index. *Nat Metab* **4**, 344–358 (2022). URL <https://www.ncbi.nlm.nih.gov/pubmed/35315439>.
- ¹¹ Couto Alves, A. *et al.* Gwas on longitudinal growth traits reveals different genetic factors influencing infant, child, and adult bmi. *Sci Adv* **5**, eaaw3095 (2019). URL <https://www.ncbi.nlm.nih.gov/pubmed/31840077>.
- ¹² Hjelmborg, J. *et al.* Genetic influences on growth traits of bmi: a longitudinal study of adult twins. *Obesity (Silver Spring)* **16**, 847–52 (2008). URL <https://www.ncbi.nlm.nih.gov/pubmed/18239571>.
- ¹³ Fabsitz, R. R., Sholinsky, P. & Carmelli, D. Genetic influences on adult weight gain and maximum body mass index in male twins. *Am J Epidemiol* **140**, 711–20 (1994). URL <https://www.ncbi.nlm.nih.gov/pubmed/7942773>.
- ¹⁴ Austin, M. A. *et al.* Genetic influences on changes in body mass index: a longitudinal analysis of women twins. *Obes Res* **5**, 326–31 (1997). URL <https://www.ncbi.nlm.nih.gov/pubmed/9285839>.
- ¹⁵ Xu, J. *et al.* Exploring the clinical and genetic associations of adult weight trajectories using electronic health records in a racially diverse biobank: a phenome-wide and polygenic risk study. *Lancet Digit Health* **4**, e604–e614 (2022). URL <https://www.ncbi.nlm.nih.gov/pubmed/35780037>.
- ¹⁶ Shilo, S., Rossman, H. & Segal, E. Axes of a revolution: challenges and promises of big data in healthcare. *Nat Med* **26**, 29–38 (2020). URL <https://www.ncbi.nlm.nih.gov/pubmed/31932803>.
- ¹⁷ Wolford, B. N., Willer, C. J. & Surakka, I. Electronic health records: the next wave of complex disease genetics. *Hum Mol Genet* **27**, R14–R21 (2018). URL <https://www.ncbi.nlm.nih.gov/pubmed/29547983>.
- ¹⁸ Wei, W. Q. & Denny, J. C. Extracting research-quality phenotypes from electronic health records to support precision medicine. *Genome Med* **7**, 41 (2015). URL <https://www.ncbi.nlm.nih.gov/pubmed/25937834>.
- ¹⁹ Gottesman, O. *et al.* The electronic medical records and genomics (emerge) network: past, present, and future. *Genet Med* **15**, 761–71 (2013). URL <https://www.ncbi.nlm.nih.gov/pubmed/23743551>.

- ²⁰ Monda, K. L. *et al.* A meta-analysis identifies new loci associated with body mass index in individuals of african ancestry. *Nat Genet* **45**, 690–6 (2013). URL <https://www.ncbi.nlm.nih.gov/pubmed/23583978>.
- ²¹ Postmus, I. *et al.* Pharmacogenetic meta-analysis of genome-wide association studies of ldl cholesterol response to statins. *Nat Commun* **5**, 5068 (2014). URL <https://www.ncbi.nlm.nih.gov/pubmed/25350695>.
- ²² Chiu, Y. F., Justice, A. E. & Melton, P. E. Longitudinal analytical approaches to genetic data. *BMC Genet* **17 Suppl 2**, 4 (2016). URL <https://www.ncbi.nlm.nih.gov/pubmed/26866891>.
- ²³ Fan, R. *et al.* Longitudinal association analysis of quantitative traits. *Genet Epidemiol* **36**, 856–69 (2012). URL <https://www.ncbi.nlm.nih.gov/pubmed/22965819>.
- ²⁴ Furlotte, N. A., Eskin, E. & Eyheramendy, S. Genome-Wide Association Mapping With Longitudinal Data. *Genetic Epidemiology* **36**, 463–471 (2012). URL <https://onlinelibrary.wiley.com/doi/abs/10.1002/gepi.21640>. _eprint: <https://onlinelibrary.wiley.com/doi/pdf/10.1002/gepi.21640>.
- ²⁵ Goldstein, J. A. *et al.* Labwas: Novel findings and study design recommendations from a meta-analysis of clinical labs in two independent biobanks. *PLoS Genet* **16**, e1009077 (2020). URL <https://www.ncbi.nlm.nih.gov/pubmed/33175840>.
- ²⁶ Justice, A. E. *et al.* Genome-wide association of trajectories of systolic blood pressure change. *BMC Proc* **10**, 321–327 (2016). URL <https://www.ncbi.nlm.nih.gov/pubmed/27980656>.
- ²⁷ Gauderman, W. J. *et al.* Longitudinal data analysis in pedigree studies. *Genet Epidemiol* **25 Suppl 1**, S18–28 (2003). URL <https://www.ncbi.nlm.nih.gov/pubmed/14635165>.
- ²⁸ Ko, S. *et al.* Gwas of longitudinal trajectories at biobank scale. *Am J Hum Genet* **109**, 433–445 (2022). URL <https://www.ncbi.nlm.nih.gov/pubmed/35196515>.
- ²⁹ Laird, N. M. & Ware, J. H. Random-Effects Models for Longitudinal Data. *Biometrics* **38**, 963–974 (1982). URL <https://www.jstor.org/stable/2529876>. Publisher: [Wiley, International Biometric Society].
- ³⁰ Xu, H. *et al.* High-throughput and efficient multilocus genome-wide association study on longitudinal outcomes. *Bioinformatics* **36**, 3004–3010 (2020). URL <https://doi.org/10.1093/bioinformatics/btaa120>.
- ³¹ Ruppert, D., Wand, M. P. & Carroll, R. J. *Semiparametric Regression*. Cambridge Series in Statistical and Probabilistic Mathematics (Cambridge University Press, Cambridge, 2003). URL <https://www.cambridge.org/core/books/semiparametric-regression/02FC9A9435232CA67532B4D31874412C>.
- ³² Das, K. *et al.* A dynamic model for genome-wide association studies. *Human Genetics* **129**, 629–639 (2011). URL <https://doi.org/10.1007/s00439-011-0960-6>.
- ³³ Das, K. *et al.* Dynamic semiparametric Bayesian models for genetic mapping of complex trait with irregular longitudinal data. *Statistics in Medicine* **32**, 509–523 (2013). URL <https://onlinelibrary.wiley.com/doi/abs/10.1002/sim.5535>. _eprint: <https://onlinelibrary.wiley.com/doi/pdf/10.1002/sim.5535>.
- ³⁴ Li, Z. & Sillanpää, M. J. A Bayesian Nonparametric Approach for Mapping Dynamic Quantitative Traits. *Genetics* **194**, 997–1016 (2013). URL <https://doi.org/10.1534/genetics.113.152736>.
- ³⁵ Li, J., Wang, Z., Li, R. & Wu, R. BAYESIAN GROUP LASSO FOR NONPARAMETRIC VARYING-COEFFICIENT MODELS WITH APPLICATION TO FUNCTIONAL GENOME-WIDE ASSOCIATION STUDIES. *The annals of applied statistics* **9**, 640–664 (2015). URL <https://www.ncbi.nlm.nih.gov/pmc/articles/PMC4605444/>.
- ³⁶ Anh Luong, D. T. & Chandola, V. A K-Means Approach to Clustering Disease Progressions. In *2017 IEEE International Conference on Healthcare Informatics (ICHI)*, 268–274 (2017).
- ³⁷ Hedman, A. K. *et al.* Identification of novel pheno-groups in heart failure with preserved ejection fraction using machine learning. *Heart* **106**, 342–349 (2020). URL <https://heart.bmj.com/content/106/5/342>. Publisher: BMJ Publishing Group Ltd and British Cardiovascular Society Section: Heart failure and cardiomyopathies.
- ³⁸ Lee, C. & Schaar, M. V. D. Temporal Phenotyping using Deep Predictive Clustering of Disease Progression. In *Proceedings of the 37th International Conference on Machine Learning*, 5767–5777 (PMLR, 2020). URL <https://proceedings.mlr.press/v119/lee20h.html>. ISSN: 2640-3498.

- ³⁹ Mullin, S. *et al.* Longitudinal K-means approaches to clustering and analyzing EHR opioid use trajectories for clinical subtypes. *Journal of Biomedical Informatics* **122**, 103889 (2021). URL <https://www.sciencedirect.com/science/article/pii/S1532046421002185>.
- ⁴⁰ Lee, C., Rashbass, J. & van der Schaar, M. Outcome-Oriented Deep Temporal Phenotyping of Disease Progression. *IEEE Transactions on Biomedical Engineering* **68**, 2423–2434 (2021). Conference Name: IEEE Transactions on Biomedical Engineering.
- ⁴¹ Carr, O., Javer, A., Rockenschaub, P., Parsons, O. & Durichen, R. Longitudinal patient stratification of electronic health records with flexible adjustment for clinical outcomes. In *Proceedings of Machine Learning for Health*, 220–238 (PMLR, 2021). URL <https://proceedings.mlr.press/v158/carr21a.html>. ISSN: 2640-3498.
- ⁴² Sudlow, C. *et al.* Uk biobank: an open access resource for identifying the causes of a wide range of complex diseases of middle and old age. *PLoS Med* **12**, e1001779 (2015). URL <https://www.ncbi.nlm.nih.gov/pubmed/25826379>.
- ⁴³ Gaziano, J. M. *et al.* Million veteran program: A mega-biobank to study genetic influences on health and disease. *J Clin Epidemiol* **70**, 214–23 (2016). URL <https://www.ncbi.nlm.nih.gov/pubmed/26441289>.
- ⁴⁴ Nguyen, X. T. *et al.* Baseline characterization and annual trends of body mass index for a mega-biobank cohort of us veterans 2011-2017. *J Health Res Rev Dev Ctries* **5**, 98–107 (2018). URL <https://www.ncbi.nlm.nih.gov/pubmed/33117892>.
- ⁴⁵ Leitsalu, L. *et al.* Cohort profile: Estonian biobank of the estonian genome center, university of tartu. *Int J Epidemiol* **44**, 1137–47 (2015). URL <https://www.ncbi.nlm.nih.gov/pubmed/24518929>.
- ⁴⁶ Pulit, S. L. *et al.* Meta-analysis of genome-wide association studies for body fat distribution in 694 649 individuals of european ancestry. *Hum Mol Genet* **28**, 166–174 (2019). URL <https://www.ncbi.nlm.nih.gov/pubmed/30239722>.
- ⁴⁷ Benonisdottir, S. *et al.* Epigenetic and genetic components of height regulation. *Nat Commun* **7**, 13490 (2016). URL <https://www.ncbi.nlm.nih.gov/pubmed/27848971>.
- ⁴⁸ Shenkman, M. *et al.* Mannosidase activity of edem1 and edem2 depends on an unfolded state of their glycoprotein substrates. *Commun Biol* **1**, 172 (2018). URL <https://www.ncbi.nlm.nih.gov/pubmed/30374462>.
- ⁴⁹ Tews, D. *et al.* Teneurin-2 (tenm2) deficiency induces ucpl expression in differentiating human fat cells. *Mol Cell Endocrinol* **443**, 106–113 (2017). URL <https://www.ncbi.nlm.nih.gov/pubmed/28088466>.
- ⁵⁰ Jung, H. *et al.* Sexually dimorphic behavior, neuronal activity, and gene expression in chd8-mutant mice. *Nat Neurosci* **21**, 1218–1228 (2018). URL <https://www.ncbi.nlm.nih.gov/pubmed/30104731>.
- ⁵¹ Mo, D. *et al.* Transcriptome landscape of porcine intramuscular adipocytes during differentiation. *J Agric Food Chem* **65**, 6317–6328 (2017). URL <https://www.ncbi.nlm.nih.gov/pubmed/28673084>.
- ⁵² Groza, T. *et al.* The international mouse phenotyping consortium: comprehensive knockout phenotyping underpinning the study of human disease. *Nucleic Acids Res* **51**, D1038–D1045 (2023). URL <https://www.ncbi.nlm.nih.gov/pubmed/36305825>.
- ⁵³ Ko, S. *et al.* GWAS of longitudinal trajectories at biobank scale. *The American Journal of Human Genetics* **109**, 433–445 (2022). URL <https://www.sciencedirect.com/science/article/pii/S0002929722000490>.
- ⁵⁴ Pirastu, N. *et al.* Genetic analyses identify widespread sex-differential participation bias. *Nat Genet* **53**, 663–671 (2021). URL <https://www.ncbi.nlm.nih.gov/pubmed/33888908>.
- ⁵⁵ Welter, D. *et al.* The nhgri gwas catalog, a curated resource of snp-trait associations. *Nucleic Acids Res* **42**, D1001–6 (2014). URL <https://www.ncbi.nlm.nih.gov/pubmed/24316577>.
- ⁵⁶ Reynolds, A. P., Richards, G., de la Iglesia, B. & Rayward-Smith, V. J. Clustering Rules: A Comparison of Partitioning and Hierarchical Clustering Algorithms. *Journal of Mathematical Modelling and Algorithms* **5**, 475–504 (2006). URL <https://doi.org/10.1007/s10852-005-9022-1>.
- ⁵⁷ Schubert, E. & Rousseeuw, P. J. Faster k-Medoids Clustering: Improving the PAM, CLARA, and CLARANS Algorithms. In Amato, G., Gennaro, C., Oria, V. & Radovanović, M. (eds.) *Similarity Search and Applications*, Lecture Notes in Computer Science, 171–187 (Springer International Publishing, Cham, 2019).

- ⁵⁸ Surakka, I. *et al.* The impact of low-frequency and rare variants on lipid levels. *Nat Genet* **47**, 589–97 (2015). URL <https://www.ncbi.nlm.nih.gov/pubmed/25961943>.
- ⁵⁹ Hoffmann, T. J. *et al.* A large electronic-health-record-based genome-wide study of serum lipids. *Nat Genet* **50**, 401–413 (2018). URL <https://www.ncbi.nlm.nih.gov/pubmed/29507422>.
- ⁶⁰ Shen, L. *et al.* Whole genome association study of brain-wide imaging phenotypes for identifying quantitative trait loci in mci and ad: A study of the adni cohort. *Neuroimage* **53**, 1051–63 (2010). URL <https://www.ncbi.nlm.nih.gov/pubmed/20100581>.
- ⁶¹ Nazarian, A., Yashin, A. I. & Kulminski, A. M. Genome-wide analysis of genetic predisposition to alzheimer's disease and related sex disparities. *Alzheimers Res Ther* **11**, 5 (2019). URL <https://www.ncbi.nlm.nih.gov/pubmed/30636644>.
- ⁶² Joshi, P. K. *et al.* Variants near chrna3/5 and apoe have age- and sex-related effects on human lifespan. *Nat Commun* **7**, 11174 (2016). URL <https://www.ncbi.nlm.nih.gov/pubmed/27029810>.
- ⁶³ Pilling, L. C. *et al.* Human longevity: 25 genetic loci associated in 389,166 uk biobank participants. *Aging (Albany NY)* **9**, 2504–2520 (2017). URL <https://www.ncbi.nlm.nih.gov/pubmed/29227965>.
- ⁶⁴ Lumsden, A. L., Mulugeta, A., Zhou, A. & Hypponen, E. Apolipoprotein e (apoe) genotype-associated disease risks: a phenome-wide, registry-based, case-control study utilising the uk biobank. *EBioMedicine* **59**, 102954 (2020). URL <https://www.ncbi.nlm.nih.gov/pubmed/32818802>.
- ⁶⁵ Astle, W. J. *et al.* The allelic landscape of human blood cell trait variation and links to common complex disease. *Cell* **167**, 1415–1429 e19 (2016). URL <https://www.ncbi.nlm.nih.gov/pubmed/27863252>.
- ⁶⁶ Kettunen, J. *et al.* Genome-wide study for circulating metabolites identifies 62 loci and reveals novel systemic effects of lpa. *Nat Commun* **7**, 11122 (2016). URL <https://www.ncbi.nlm.nih.gov/pubmed/27005778>.
- ⁶⁷ Shrine, N. *et al.* New genetic signals for lung function highlight pathways and chronic obstructive pulmonary disease associations across multiple ancestries. *Nat Genet* **51**, 481–493 (2019). URL <https://www.ncbi.nlm.nih.gov/pubmed/30804560>.
- ⁶⁸ Bulik-Sullivan, B. K. *et al.* Ld score regression distinguishes confounding from polygenicity in genome-wide association studies. *Nat Genet* **47**, 291–5 (2015). URL <https://www.ncbi.nlm.nih.gov/pubmed/25642630>.
- ⁶⁹ International HapMap, C. *et al.* Integrating common and rare genetic variation in diverse human populations. *Nature* **467**, 52–8 (2010). URL <https://www.ncbi.nlm.nih.gov/pubmed/20811451>.
- ⁷⁰ Song, M. *et al.* Associations between genetic variants associated with body mass index and trajectories of body fatness across the life course: a longitudinal analysis. *Int J Epidemiol* **47**, 506–515 (2018). URL <https://www.ncbi.nlm.nih.gov/pubmed/29211904>.
- ⁷¹ Bray, M. S. *et al.* Nih working group report-using genomic information to guide weight management: From universal to precision treatment. *Obesity (Silver Spring)* **24**, 14–22 (2016). URL <https://www.ncbi.nlm.nih.gov/pubmed/26692578>.
- ⁷² Delahanty, L. M. *et al.* Genetic predictors of weight loss and weight regain after intensive lifestyle modification, metformin treatment, or standard care in the diabetes prevention program. *Diabetes Care* **35**, 363–6 (2012). URL <https://www.ncbi.nlm.nih.gov/pubmed/22179955>.
- ⁷³ Delahanty, L. M. *et al.* Genetic predictors of weight loss and weight regain after intensive lifestyle modification, metformin treatment, or standard care in the diabetes prevention program. *Diabetes Care* **35**, 363–6 (2012). URL <https://www.ncbi.nlm.nih.gov/pubmed/22179955>.
- ⁷⁴ Liou, T. H. *et al.* Esr1, fto, and ucp2 genes interact with bariatric surgery affecting weight loss and glycemic control in severely obese patients. *Obes Surg* **21**, 1758–65 (2011). URL <https://www.ncbi.nlm.nih.gov/pubmed/21720911>.
- ⁷⁵ Sarzynski, M. A. *et al.* Associations of markers in 11 obesity candidate genes with maximal weight loss and weight regain in the sos bariatric surgery cases. *Int J Obes (Lond)* **35**, 676–83 (2011). URL <https://www.ncbi.nlm.nih.gov/pubmed/20733583>.

- ⁷⁶ Zhang, X. *et al.* Fto genotype and 2-year change in body composition and fat distribution in response to weight-loss diets: the pounds lost trial. *Diabetes* **61**, 3005–11 (2012). URL <https://www.ncbi.nlm.nih.gov/pubmed/22891219>.
- ⁷⁷ Papandonatos, G. D. *et al.* Genetic predisposition to weight loss and regain with lifestyle intervention: Analyses from the diabetes prevention program and the look ahead randomized controlled trials. *Diabetes* **64**, 4312–21 (2015). URL <https://www.ncbi.nlm.nih.gov/pubmed/26253612>.
- ⁷⁸ McCaffery, J. M. *et al.* Genetic predictors of change in waist circumference and waist-to-hip ratio with lifestyle intervention: The trans-nih consortium for genetics of weight loss response to lifestyle intervention. *Diabetes* **71**, 669–676 (2022). URL <https://www.ncbi.nlm.nih.gov/pubmed/35043141>.
- ⁷⁹ Holzapfel, C. *et al.* Association between single nucleotide polymorphisms and weight reduction in behavioural interventions—a pooled analysis. *Nutrients* **13** (2021). URL <https://www.ncbi.nlm.nih.gov/pubmed/33801339>.
- ⁸⁰ Nelson, M. R. *et al.* The support of human genetic evidence for approved drug indications. *Nat Genet* **47**, 856–60 (2015). URL <https://www.ncbi.nlm.nih.gov/pubmed/26121088>.
- ⁸¹ Silventoinen, K. & Kaprio, J. Genetics of tracking of body mass index from birth to late middle age: evidence from twin and family studies. *Obes Facts* **2**, 196–202 (2009). URL <https://www.ncbi.nlm.nih.gov/pubmed/20054225>.
- ⁸² Winkler, T. W. *et al.* The Influence of Age and Sex on Genetic Associations with Adult Body Size and Shape: A Large-Scale Genome-Wide Interaction Study. *PLoS Genetics* **11**, e1005378 (2015). URL <https://journals.plos.org/plosgenetics/article?id=10.1371/journal.pgen.1005378>. Publisher: Public Library of Science.
- ⁸³ Gillespie, N. A. *et al.* Determining the stability of genome-wide factors in BMI between ages 40 to 69 years. *PLoS Genetics* **18**, e1010303 (2022). URL <https://journals.plos.org/plosgenetics/article?id=10.1371/journal.pgen.1010303>. Publisher: Public Library of Science.
- ⁸⁴ Beesley, L. J., Fritsche, L. G. & Mukherjee, B. A modeling framework for exploring sampling and observation process biases in genome and phenome-wide association studies using electronic health records. *bioRxiv* (2019). URL <https://www.biorxiv.org/content/early/2019/05/14/499392>. <https://www.biorxiv.org/content/early/2019/05/14/499392.full.pdf>.
- ⁸⁵ Fry, A. *et al.* Comparison of sociodemographic and health-related characteristics of uk biobank participants with those of the general population. *Am J Epidemiol* **186**, 1026–1034 (2017). URL <https://www.ncbi.nlm.nih.gov/pubmed/28641372>.
- ⁸⁶ Goudie, R. J. B., Presanis, A. M., Lunn, D., Angelis, D. D. & Wernisch, L. Joining and Splitting Models with Markov Melding. *Bayesian Analysis* **14**, 81–109 (2019). URL <https://projecteuclid.org/journals/bayesian-analysis/volume-14/issue-1/Joining-and-Splitting-Models-with-Markov-Melding/10.1214/18-BA1104.full>. Publisher: International Society for Bayesian Analysis.
- ⁸⁷ Loh, P. R. *et al.* Efficient bayesian mixed-model analysis increases association power in large cohorts. *Nat Genet* **47**, 284–90 (2015). URL <https://www.ncbi.nlm.nih.gov/pubmed/25642633>.
- ⁸⁸ Li, H. *et al.* Triglyceride-glucose index variability and incident cardiovascular disease: a prospective cohort study. *Cardiovasc Diabetol* **21**, 105 (2022). URL <https://www.ncbi.nlm.nih.gov/pubmed/35689232>.
- ⁸⁹ Nuyujukian, D. S. *et al.* Blood pressure variability and risk of heart failure in accord and the vadt. *Diabetes Care* **43**, 1471–1478 (2020). URL <https://www.ncbi.nlm.nih.gov/pubmed/32327422>.
- ⁹⁰ Speakman, J. R. *et al.* Set points, settling points and some alternative models: theoretical options to understand how genes and environments combine to regulate body adiposity. *Dis Model Mech* **4**, 733–45 (2011). URL <https://www.ncbi.nlm.nih.gov/pubmed/22065844>.
- ⁹¹ Muller, M. J., Geisler, C., Heymsfield, S. B. & Bosy-Westphal, A. Recent advances in understanding body weight homeostasis in humans. *F1000Res* **7** (2018). URL <https://www.ncbi.nlm.nih.gov/pubmed/30026913>.
- ⁹² Nawaz, H., Chan, W., Abdulrahman, M., Larson, D. & Katz, D. L. Self-reported weight and height: implications for obesity research. *Am J Prev Med* **20**, 294–8 (2001). URL <https://www.ncbi.nlm.nih.gov/pubmed/11331120>.

- ⁹³ Kowal, R. C., Herz, J., Goldstein, J. L., Esser, V. & Brown, M. S. Low density lipoprotein receptor-related protein mediates uptake of cholesteryl esters derived from apoprotein e-enriched lipoproteins. *Proc Natl Acad Sci U S A* **86**, 5810–4 (1989). URL <https://www.ncbi.nlm.nih.gov/pubmed/2762297>.
- ⁹⁴ Kockx, M., Traini, M. & Kritharides, L. Cell-specific production, secretion, and function of apolipoprotein e. *J Mol Med (Berl)* **96**, 361–371 (2018). URL <https://www.ncbi.nlm.nih.gov/pubmed/29516132>.
- ⁹⁵ Garrison, R. J. *et al.* Obesity and lipoprotein cholesterol in the framingham offspring study. *Metabolism* **29**, 1053–60 (1980). URL <https://www.ncbi.nlm.nih.gov/pubmed/7432169>.
- ⁹⁶ Albrink, M. J. *et al.* Intercorrelations among plasma high density lipoprotein, obesity and triglycerides in a normal population. *Lipids* **15**, 668–76 (1980). URL <https://www.ncbi.nlm.nih.gov/pubmed/7421421>.
- ⁹⁷ Panagiotakos, D. B., Pitsavos, C., Yannakoulia, M., Chrysohoou, C. & Stefanadis, C. The implication of obesity and central fat on markers of chronic inflammation: The attica study. *Atherosclerosis* **183**, 308–15 (2005). URL <https://www.ncbi.nlm.nih.gov/pubmed/16285994>.
- ⁹⁸ Purdy, J. C. & Shatzel, J. J. The hematologic consequences of obesity. *Eur J Haematol* **106**, 306–319 (2021). URL <https://www.ncbi.nlm.nih.gov/pubmed/33270290>.
- ⁹⁹ Gillette Guyonnet, S. *et al.* Iana (international academy on nutrition and aging) expert group: weight loss and alzheimer's disease. *J Nutr Health Aging* **11**, 38–48 (2007). URL <https://www.ncbi.nlm.nih.gov/pubmed/17315079>.
- ¹⁰⁰ von Hardenberg, S., Gnewuch, C., Schmitz, G. & Borlak, J. Apoe is a major determinant of hepatic bile acid homeostasis in mice. *J Nutr Biochem* **52**, 82–91 (2018). URL <https://www.ncbi.nlm.nih.gov/pubmed/29175670>.
- ¹⁰¹ Wang, J. *et al.* Apoe and the role of very low density lipoproteins in adipose tissue inflammation. *Atherosclerosis* **223**, 342–9 (2012). URL <https://www.ncbi.nlm.nih.gov/pubmed/22770993>.
- ¹⁰² Blanchard, J. W. *et al.* Apoe4 impairs myelination via cholesterol dysregulation in oligodendrocytes. *Nature* **611**, 769–779 (2022). URL <https://www.ncbi.nlm.nih.gov/pubmed/36385529>.
- ¹⁰³ Greendale, G. A. *et al.* Changes in body composition and weight during the menopause transition. *JCI Insight* **4** (2019). URL <https://www.ncbi.nlm.nih.gov/pubmed/30843880>.
- ¹⁰⁴ Davies, K. M., Heaney, R. P., Recker, R. R., Barger-Lux, M. J. & Lappe, J. M. Hormones, weight change and menopause. *Int J Obes Relat Metab Disord* **25**, 874–9 (2001). URL <https://www.ncbi.nlm.nih.gov/pubmed/11439302>.
- ¹⁰⁵ Chen, Y. W., Hang, D., Kvaerner, A. S., Giovannucci, E. & Song, M. Associations between body shape across the life course and adulthood concentrations of sex hormones in men and pre- and postmenopausal women: a multicohort study. *Br J Nutr* **127**, 1000–1009 (2022). URL <https://www.ncbi.nlm.nih.gov/pubmed/34187605>.
- ¹⁰⁶ Conroy, M. *et al.* The advantages of uk biobank's open-access strategy for health research. *J Intern Med* **286**, 389–397 (2019). URL <https://www.ncbi.nlm.nih.gov/pubmed/31283063>.
- ¹⁰⁷ Coady, S. A. *et al.* Genetic variability of adult body mass index: a longitudinal assessment in framingham families. *Obes Res* **10**, 675–81 (2002). URL <https://www.ncbi.nlm.nih.gov/pubmed/12105290>.
- ¹⁰⁸ Singh, P. *et al.* Statins decrease leptin expression in human white adipocytes. *Physiol Rep* **6** (2018). URL <https://www.ncbi.nlm.nih.gov/pubmed/29372612>.
- ¹⁰⁹ McCarron, D. A. & Reusser, M. E. Body weight and blood pressure regulation. *Am J Clin Nutr* **63**, 423S–425S (1996). URL <https://www.ncbi.nlm.nih.gov/pubmed/8615333>.
- ¹¹⁰ Hernan, M. A., Hernandez-Diaz, S. & Robins, J. M. A structural approach to selection bias. *Epidemiology* **15**, 615–25 (2004). URL <https://www.ncbi.nlm.nih.gov/pubmed/15308962>.
- ¹¹¹ Beesley, L. J. *et al.* The emerging landscape of health research based on biobanks linked to electronic health records: Existing resources, statistical challenges, and potential opportunities. *Stat Med* **39**, 773–800 (2020). URL <https://www.ncbi.nlm.nih.gov/pubmed/31859414>.
- ¹¹² Kutcher, S. A., Brophy, J. M., Banack, H. R., Kaufman, J. S. & Samuel, M. Emulating a randomised controlled trial with observational data: An introduction to the target trial framework. *Can J Cardiol* **37**, 1365–1377 (2021). URL <https://www.ncbi.nlm.nih.gov/pubmed/34090982>.

- ¹¹³ Shortreed, S. M., Rutter, C. M., Cook, A. J. & Simon, G. E. Improving pragmatic clinical trial design using real-world data. *Clin Trials* **16**, 273–282 (2019). URL <https://www.ncbi.nlm.nih.gov/pubmed/30866672>.
- ¹¹⁴ Bycroft, C. *et al.* The uk biobank resource with deep phenotyping and genomic data. *Nature* **562**, 203–209 (2018). URL <https://www.ncbi.nlm.nih.gov/pubmed/30305743>.
- ¹¹⁵ Team, U. B. *UK Biobank Primary Care Linked Data* (2019), version 1.0 edn. URL https://biobank.ndph.ox.ac.uk/showcase/showcase/docs/primary_care_data.pdf.
- ¹¹⁶ Kuan, V. *et al.* A chronological map of 308 physical and mental health conditions from 4 million individuals in the english national health service. *Lancet Digit Health* **1**, e63–e77 (2019). URL <https://www.ncbi.nlm.nih.gov/pubmed/31650125>.
- ¹¹⁷ Bates, D., Machler, M., Bolker, B. & S., W. Fitting linear mixed-effects models using lme4. *Journal of Statistical Software* **67**, 1–48 (2015). URL <https://doi.org/10.18637/jss.v067.i01>.
- ¹¹⁸ R Core Team. *R: A Language and Environment for Statistical Computing* (R Foundation for Statistical Computing, Vienna, Austria, 2021). URL <https://www.R-project.org/>.
- ¹¹⁹ Beasley, T. M., Erickson, S. & Allison, D. B. Rank-Based Inverse Normal Transformations are Increasingly Used, But are They Merited? *Behavior Genetics* **39**, 580–595 (2009). URL <https://doi.org/10.1007/s10519-009-9281-0>.
- ¹²⁰ Eilers, P. H. C. & Marx, B. D. Flexible smoothing with B-splines and penalties. *Statistical Science* **11**, 89–121 (1996). URL <https://projecteuclid.org/journals/statistical-science/volume-11/issue-2/Flexible-smoothing-with-B-splines-and-penalties/10.1214/ss/1038425655.full>. Publisher: Institute of Mathematical Statistics.
- ¹²¹ O’Hagan, A. & Kendall, M. G. *Kendall’s Advanced Theory of Statistics: Bayesian inference. Volume 2B* (Edward Arnold, 1994). Google-Books-ID: DlrEMgEACAAJ.
- ¹²² Maechler, M., Rousseeuw, P., Struyf, A., Hubert, M. & Hornik, K. *cluster: Cluster Analysis Basics and Extensions* (2022). URL <https://CRAN.R-project.org/package=cluster>. R package version 2.1.4 — For new features, see the ‘Changelog’ file (in the package source).
- ¹²³ Peter, J. R. Silhouettes: A graphical aid to the interpretation and validation of cluster analysis. *Journal of Computational and Applied Mathematics* **20**, 53–65 (1987). URL <https://www.sciencedirect.com/science/article/pii/0377042787901257>.
- ¹²⁴ Smithson, M. & Verkuilen, J. A better lemon squeezer? maximum-likelihood regression with beta-distributed dependent variables. *Psychol Methods* **11**, 54–71 (2006). URL <https://www.ncbi.nlm.nih.gov/pubmed/16594767>.
- ¹²⁵ Benner, C. *et al.* Finemap: efficient variable selection using summary data from genome-wide association studies. *Bioinformatics* **32**, 1493–501 (2016). URL <https://www.ncbi.nlm.nih.gov/pubmed/26773131>.
- ¹²⁶ Yang, J., Lee, S. H., Goddard, M. E. & Visscher, P. M. Gcta: a tool for genome-wide complex trait analysis. *Am J Hum Genet* **88**, 76–82 (2011). URL <https://www.ncbi.nlm.nih.gov/pubmed/21167468>.
- ¹²⁷ Chang, C. C. *et al.* Second-generation plink: rising to the challenge of larger and richer datasets. *Gigascience* **4**, 7 (2015). URL <https://www.ncbi.nlm.nih.gov/pubmed/25722852>.
- ¹²⁸ Venables, W. N. & Ripley, B. D. *Modern Applied Statistics with S* (Springer, New York, 2002), fourth edn. URL <https://www.stats.ox.ac.uk/pub/MASS4/>. ISBN 0-387-95457-0.
- ¹²⁹ Hunter-Zinck, H. *et al.* Genotyping array design and data quality control in the million veteran program. *Am J Hum Genet* **106**, 535–548 (2020). URL <https://www.ncbi.nlm.nih.gov/pubmed/32243820>.
- ¹³⁰ Mbatchou, J. *et al.* Computationally efficient whole-genome regression for quantitative and binary traits. *Nature Genetics* **53**, 1097–1103 (2021). URL <https://www.nature.com/articles/s41588-021-00870-7>. Number: 7 Publisher: Nature Publishing Group.
- ¹³¹ Mitt, M. *et al.* Improved imputation accuracy of rare and low-frequency variants using population-specific high-coverage wgs-based imputation reference panel. *Eur J Hum Genet* **25**, 869–876 (2017). URL <https://www.ncbi.nlm.nih.gov/pubmed/28401899>.

- ¹³² Palmer, C. & Pe'er, I. Statistical correction of the winner's curse explains replication variability in quantitative trait genome-wide association studies. *PLoS Genet* **13**, e1006916 (2017). URL <https://www.ncbi.nlm.nih.gov/pubmed/28715421>.
- ¹³³ Denaxas, S. *et al.* A semi-supervised approach for rapidly creating clinical biomarker phenotypes in the uk biobank using different primary care ehr and clinical terminology systems. *JAMIA Open* **3**, 545–556 (2020). URL <https://www.ncbi.nlm.nih.gov/pubmed/33619467>.
- ¹³⁴ Bulik-Sullivan, B. *et al.* An atlas of genetic correlations across human diseases and traits. *Nat Genet* **47**, 1236–41 (2015). URL <https://www.ncbi.nlm.nih.gov/pubmed/26414676>.
- ¹³⁵ Genomes Project, C. *et al.* A global reference for human genetic variation. *Nature* **526**, 68–74 (2015). URL <https://www.ncbi.nlm.nih.gov/pubmed/26432245>.
- ¹³⁶ Bezanson, J., Edelman, A., Karpinski, S. & Shah, V. B. Julia: A fresh approach to numerical computing. *SIAM Review* **59**, 65–98 (2017). URL <https://doi.org/10.1137/141000671>.
- ¹³⁷ Loos, R. J. & Yeo, G. S. The bigger picture of fto: the first gwas-identified obesity gene. *Nat Rev Endocrinol* **10**, 51–61 (2014). URL <https://www.ncbi.nlm.nih.gov/pubmed/24247219>.

REVIEWERS' COMMENTS

Reviewer #1 (Remarks to the Author):

All my previous questions and comments have been adequately addressed. I do not have any further suggestions.

Reviewer #1 (Remarks on code availability):

The code provided in the manuscript is not accessible,
https://github.com/lindgrengroup/longitudinal_primarycare/tree/main/adiposity/scripts/manuscript

Reviewer #2 (Remarks to the Author):

I'd like to congratulate to the authors for their very thorough revision. All my comments have been addressed. The replication analysis in MVP and in the Estonian Biobank is very neat. On the other hand, I could not find the code on GitHub and the link for the code provided in the paper was not found.

Reviewer #2 (Remarks on code availability):

Code not available on GitHub.

Reviewer #3 (Remarks to the Author):

Authors addressed all the comments

REVIEWERS' COMMENTS

Reviewer #1 (Remarks to the Author):

All my previous questions and comments have been adequately addressed. I do not have any further suggestions.

Authors' reply: Thank you for your valuable feedback that has greatly improved our work!

Reviewer #1 (Remarks on code availability):

The code provided in the manuscript is not accessible, https://github.com/lindgrengroup/longitudinal_primarycare/tree/main/adiposity/scripts/manuscript

Authors' reply: We have made this publicly accessible now that the manuscript has been accepted in principle.

Reviewer #2 (Remarks to the Author):

I'd like to congratulate to the authors for their very thorough revision. All my comments have been addressed. The replication analysis in MVP and in the Estonian Biobank is very neat. On the other hand, I could not find the code on GitHub and the link for the code provided in the paper was not found.

Authors' reply: Thank you for your generous feedback! We have made the code publicly accessible now that the manuscript has been accepted in principle.

Reviewer #2 (Remarks on code availability):

Code not available on GitHub.

Authors' reply: Thank you for your generous feedback! We have made the code publicly accessible now that the manuscript has been accepted in principle.

Reviewer #3 (Remarks to the Author):

Authors addressed all the comments

Authors' reply: Thank you for your time in reviewing the work.